# Growth phase estimation for abundant bacterial populations sampled longitudinally from human stool metagenomes

Joe J. Lim [1], Christian Diener [2], James Wilson [2], Jacob J. Valenzuela[2], Nitin S. Baliga[2,3,4,5,6] & Sean M. Gibbons [2,6,7,8,9] ✉

Longitudinal sampling of the stool has yielded important insights into the ecological dynamics of the human gut microbiome. However, human stool samples are available approximately once per day, while commensal population doubling times are likely on the order of minutes-to-hours. Despite this mismatch in timescales, much of the prior work on human gut microbiome time series modeling has assumed that day-to-day fluctuations in taxon abundances are related to population growth or death rates, which is likely not the case. Here, we propose an alternative model of the human gut as a stationary system, where population dynamics occur internally and the bacterial population sizes measured in a bolus of stool represent a steady-state endpoint of these dynamics. We formalize this idea as stochastic logistic growth. We show how this model provides a path toward estimating the growth phases of gut bacterial populations in situ. We validate our model predictions using an in vitro *Escherichia coli* growth experiment. Finally, we show how this method can be applied to densely-sampled human stool metagenomic time series data. We discuss how these growth phase estimates may be used to better inform metabolic modeling in flow-through ecosystems, like animal guts or industrial bioreactors.

The human gut is an anaerobic bioreactor, ecologically distinct to each individual, that transforms dietary and host substrates into bioactive molecules important to host health[1–3]. Disruptions to the ecological composition of the gut have been shown to mediate the progression of various diseases[4–8]. Furthermore, the ecological dynamics of the gut appear to be relevant to both health and disease states[9,10]. However, the biological interpretation of densely-sampled adult human fecal microbiome time series is fraught.

Various dynamical models have been applied to gut microbial abundance data collected from adult human donors[11–15]. These models

often assume, either explicitly or implicitly, that day-to-day changes in abundance are proportional to population growth and/or death[16]. However, the underlying data often do not match this assumption[11,16–20]. The gut is a flow-through ecosystem and commensal gut bacteria must grow fast enough to avoid dilution-to-extinction. As such, gut bacterial doubling times tend to be fast, likely ranging from minutes-to-hours, although precise in vivo estimates are not available (we contend that doubling times of a day or more in the gut would not be sufficient to maintain a stable population size with a daily defecation rate)[21–23]. However, stool sampling frequency is usually limited to,

[1]Department of Environmental & Occupational Health Sciences, University of Washington, Seattle, WA 98105, USA. [2]Institute for Systems Biology, Seattle, WA 98109, USA. [3]Departments of Biology and Microbiology, University of Washington, Seattle, WA 98105, USA. [4]Lawrence Berkeley National Laboratory, CA 94720 Berkeley, USA. [5]Molecular and Cellular Biology Program, University of Washington, WA 98105 Seattle, USA. [6]Molecular Engineering Graduate Program, University of Washington, WA 98105 Seattle, USA. [7]Department of Bioengineering, University of Washington, Seattle, WA 98105, USA. [8]Department of Genome Sciences, University of Washington, Seattle, WA 98105, USA. [9]eScience Institute, University of Washington, Seattle, WA 98105, USA. ✉e-mail: sgibbons@isbscience.org

at most, about once per day. Consequently, rapid internal population dynamics likely cannot be directly estimated from the day-to-day measurements obtained from stool[16].

Given these sampling limitations, and in the absence of major perturbations that require multi-day recovery processes in the human gut, it is unclear whether or not meaningful insights into commensal population dynamics can be gleaned from adult human gut microbiome time series. One workaround for inferring effective growth rates of bacterial populations in situ is to leverage metagenome-inferred replication rates[21,22]. Briefly, instantaneous replication rates can be estimated for abundant bacterial populations in metagenomic samples by taking advantage of the fact that fast-growing taxa show an asymmetry in reads mapping to different genomic loci, with higher read depth near the origin of replication and a lower depth near the terminus due to the initiation of multiple replication forks[21–23]. However, even when replication rates and population abundances can both be estimated from the same metagenomic samples, it is unclear how these measurements are related to the in situ growth phase of a population.

Early experiments by Jaques Monod[24] identified distinct growth phases for bacterial populations in culture, which can be captured by the stochastic logistic growth equation (sLGE)[25]. The sLGE has been shown to be a good fit for bacterial population growth in vitro and in real-world, steady-state ecosystems[26–32]. We used the sLGE to study statistical relationships between population sizes and growth rates across the various phases of growth (i.e., acceleration, mid-log, deceleration, and stationary phases) to see if we could extract in situ growth phase information. Overall, the sLGE model yields statistical relationships that may be leveraged to identify the in situ growth phase of a bacterial population sampled at a regular period from a quasi-batch-culture, flow-through, steady-state ecosystem, like the human gut.

To assess our model predictions, we sampled *Eschericial coli* populations at different points along the growth curve. We calculated population sizes and replication rates for these samples and observed excellent agreement between this in vitro model and our sLGE simulations. We also measured population abundance and replication rate trajectories from more than a dozen organisms across four densely sampled human gut metagenomic time series[33]. On average, when controlling for taxonomy, gut commensal growth rates and population sizes were positively correlated, both cross-sectionally over 84 stool donors and longitudinally within each of four stool donor time series, which suggests that most abundant taxa in the gut are growing exponentially when sampled in stool. However, we saw more heterogeneity for specific taxa within individual donor time series. We were able to identify specific growth phase signatures in abundant bacterial populations in the guts of four individuals with long and dense metagenomic time series by analyzing paired replication rate and abundance trajectories. We describe how our growth phase inference approach can serve to improve statistical inferences derived from microbiome data and to inform more accurate mechanistic modeling of flow-through ecosystems (e.g., community-scale metabolic models, which usually assume exponential growth), which could have broad implications for human health[8,34,35], agricultural systems[36,37], climate change[36,38,39], and industrial bioreactors[40,41].

## Results

### Framing the gut as an anaerobic flow-through bioreactor

The mammalian gut can be understood as an anaerobic batch culture reactor with a semi-continuous input (i.e., discrete boluses of dietary inputs, mixed with host substrates like mucin and bile acids) and output (i.e., discrete boluses of stool)[42], and microbial taxa must grow fast enough within the system to avoid dilution-to-extinction (Fig. 1a). Thus, stool sampling captures the endpoint of internal gut bacterial population dynamics. For example, in our conceptual figure we see

that Taxon 1 starts growing higher up in the colon and is in stationary phase by the time a stool sample is collected, while Taxon 3 starts growing lower in the colon and is still growing exponentially at the point of stool sampling (Fig. 1a). Overall, the daily abundances of Taxa 1–3 represent the average ($\mu$) steady-state population size, plus or minus some amount of biological and technical noise, at the time of stool sampling (Fig. 1a). To investigate improved methods for interpreting the dynamics of human gut microbial time series, we downloaded shotgun metagenomic time series data from the BIO-ML cohort (i.e., health-screened stool donors who provided fecal-transplant material to the stool bank OpenBiome)[33]. The BIO-ML cohort contained 84 donors[33]. To filter for dense longitudinal data, we selected a subset of donors with more than 50 time points. Four donors (i.e., donors ae, am, an, and ao) met this criterion, with 3–5 fecal samples per week for >50 days (Fig. 1b).

### Characterizing the relationships between gut commensal population size and growth rate using metagenomic time series data

We first investigated the statistical properties of day-to-day fluctuations in gut bacterial population sizes, estimated from fecal shotgun metagenomic time series. Specifically, we looked at the associations between population abundance estimates ($t_n$) and the changes in abundance estimates (i.e., deltas) between time points ($t_{n+1} - t_n$). Naïvely, if most bacterial populations in stool were growing exponentially, we would expect that population abundances and growth rates would be positively correlated. However, prior work has indicated an overall negative correlation between abundances and changes in abundances in stool 16 S rRNA gene amplicon sequencing data generated from densely sampled human stool time series[15,33]. Indeed, we found that abundant bacterial populations in the stool of the four BIO-ML donors maintained stable average abundances over time ($\mu$), with day-to-day fluctuations above and below this average, as pictured in the example of *Bacteroides cellulosilyticus* in donor am (Fig. 2a, b). This kind of pattern mirrors what one would expect when randomly sampling from a stationary distribution (Fig. 2b). We observed that the deltas ($t_{n+1} - t_n$) for the same gut taxon (*Bacteroides uniformis*) measured across each donor time series, when plotted against their respective normalized abundances ($t_n$), showed the same negative association (Fig. 2c). Furthermore, similar negative associations were uniformly observed across all taxa analyzed, across all four donors (Fig. 2d). This consistent negative association between population abundances and changes in abundance between time points is strongly consistent with sampling from a stationary distribution, which is equivalent to 'regression-to-the-mean' as an organism fluctuates around a fixed carrying capacity, similar to what we have reported previously[15,32].

One important ecological factor that can impact gut microbial dynamics is host diet[43,44]. Although changes in dietary intake can alter microbial abundance, average dietary choices are highly conserved within an individual and these choices are notoriously difficult to modify outside of radical changes in geography or lifestyle[45–47]. Prior work demonstrated that macronutrient intake within an individual is largely stable over time and does not show significant autocorrelation or drift[15,48]. Indeed, for donor A from this prior study, we found that longitudinal measurements of macronutrients (i.e., daily intake of calories, carbohydrates, protein, fat, fiber, cholesterol, saturated fat, sugar, sodium, calcium) were stationary over several months, despite day-to-day fluctuations (Fig. S1). Combined with the overwhelming stationarity of microbial abundance trajectories within healthy individuals not undergoing major lifestyle changes[15,32,33], these results support our assertion that dietary patterns are largely stable over weeks-to-months and stool samples provide stable, steady-state population abundance estimates of abundant gut commensal bacteria. Furthermore, we have shown in prior studies on these same BIO-ML

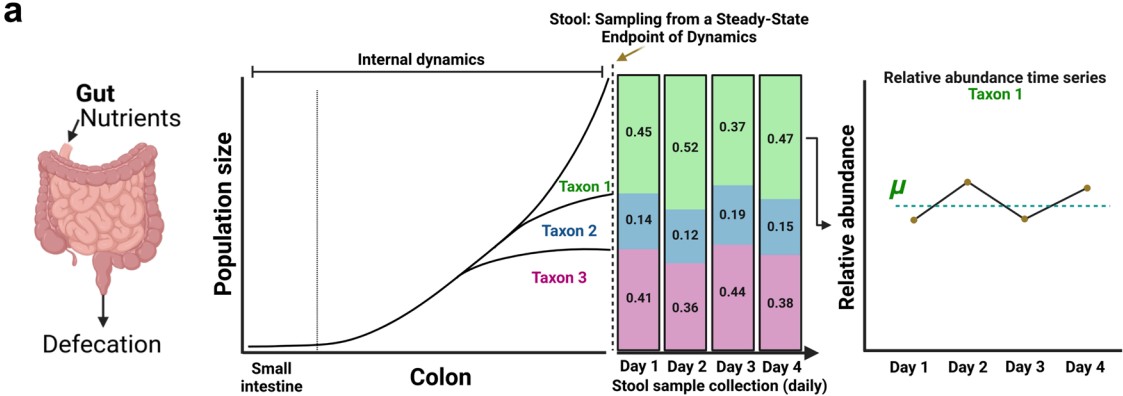

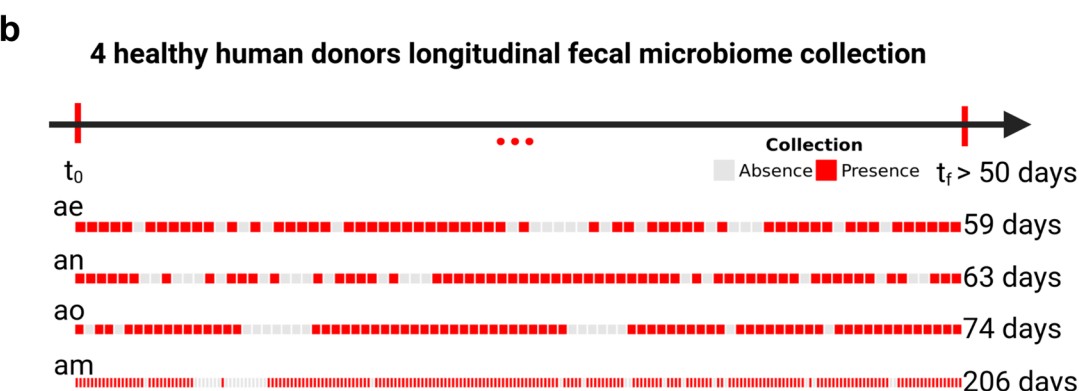

**Fig. 1 | Conceptual figure showing internal bacterial population dynamics within a human gut and the observed steady-state abundance derived from stool sampling. a** Mammalian guts are continuous flow-through systems. Taxa grow in the large intestine with varying growth rates, carrying capacities, and steady-state population sizes, and may be in different growth phases at the time of stool sampling. For example, see dynamics for Taxa 1–3. Daily stool collections show variation in abundances, but this variation likely does not reflect internal growth dynamics in the gut. **b** Healthy BIO-ML stool donors (subject IDs: ae, am, an, and ao) with samples collected 3-5 days per week for a total of >50 time points. Red indicates presence of shotgun metagenomic sequencing data and gray represents absence of metagenomic data from consecutive daily time points.

participants that average within-person taxon abundances are highly correlated across people, which suggests that these carrying capacities, likely representing dietary and host substrate niches, are fairly conserved across humans[15,33].

Next, we looked at the statistical associations between calculated peak-to-trough ratios (i.e., PTRs; a proxy for the effective growth-rate) for abundant bacterial populations from each metagenomic sample and their respective metagenomic population abundance estimates[22]. If the deltas, presented above, were truly proportional to growth and/ or death rates, we would expect that the statistical relationships between deltas and population size would be similar to those between PTRs and population size. However, unlike the putative regression-to-the-mean signature identified for the deltas, we found variable statistical relationships between $\log_2$(PTR) and centered log-ratio (CLR) transformed population abundances for the same taxon across the four donors (*Bacteroides ovatus_1*, Fig. 3a). Similarly, we saw a wide range of positive, negative, and null associations between $\log_2$(PTRs) and CLR abundances across all measured taxa within each donor (Fig. 3b). These results are inconsistent with a regression-to-the-mean signal, and suggest a more complex relationship between growth rate and population size[49–51]. Finally, we calculated temporally-averaged PTRs and population sizes for each abundant taxon within each of the four donors. Overall, there was a significantly positive association (linear regression, *p* values = 0.0318, 0.125, 0.155, 0.031 for donors ae, am, an, and ao, respectively; combined *p-value* using Fisher's method = 0.005) between average $\log_2$(PTR) and average CLR abundance across all four donors (Fig. 3c), indicating that taxa with higher average

population sizes tend to have higher average growth rates. We also looked into whether or not $\log_2$(PTR) magnitudes were inter-comparable across taxa (Fig. S2). We calculated $\log_2$(PTRs) for all abundant taxa detected across all 84 BIO-ML donors and found that the median $\log_2$(PTR) was fairly similar across taxonomic classes (-0.45–0.75), with most classes showing a wide range (Fig. S2). To assess whether or not $\log_2$(PTR)-CLR associations were robust to controlling for taxonomy, we included either class- or species-level categorizations as covariates in a linear regression model and saw a significant positive association, independent of taxonomy (class-level $\beta = 0.0612$, $p = 8.359e^{-60}$; species-level $\beta = 0.0101$, $p = 0.0006$). While we saw a significantly positive cross-sectional association between PTRs and abundances when controlling for species identity, we found that the vast majority of individual species showed null associations between abundance-PTR relationships, indicating that we were perhaps underpowered to detect this weak effect at the level of individual taxa (Figs. 3 and S3). For three species, we actually saw significant negative associations: *Alistipes shaii*, *Alistipes finegoildii*, and *Odoribacter splanchnicus* (FDR-adjusted $p < 0.1$; Fig. S3).

**Stochastic logistic growth equation provides insights into growth phases**

In order to better understand and interpret the varying relationships we observe between $\log_2$(PTRs) and CLR abundance time series, we used a modeling approach. The basic properties of growth curves of microbial taxa can be captured using the logistic growth equation (LGE) (Fig. 4). Although simple models like the LGE do not capture the

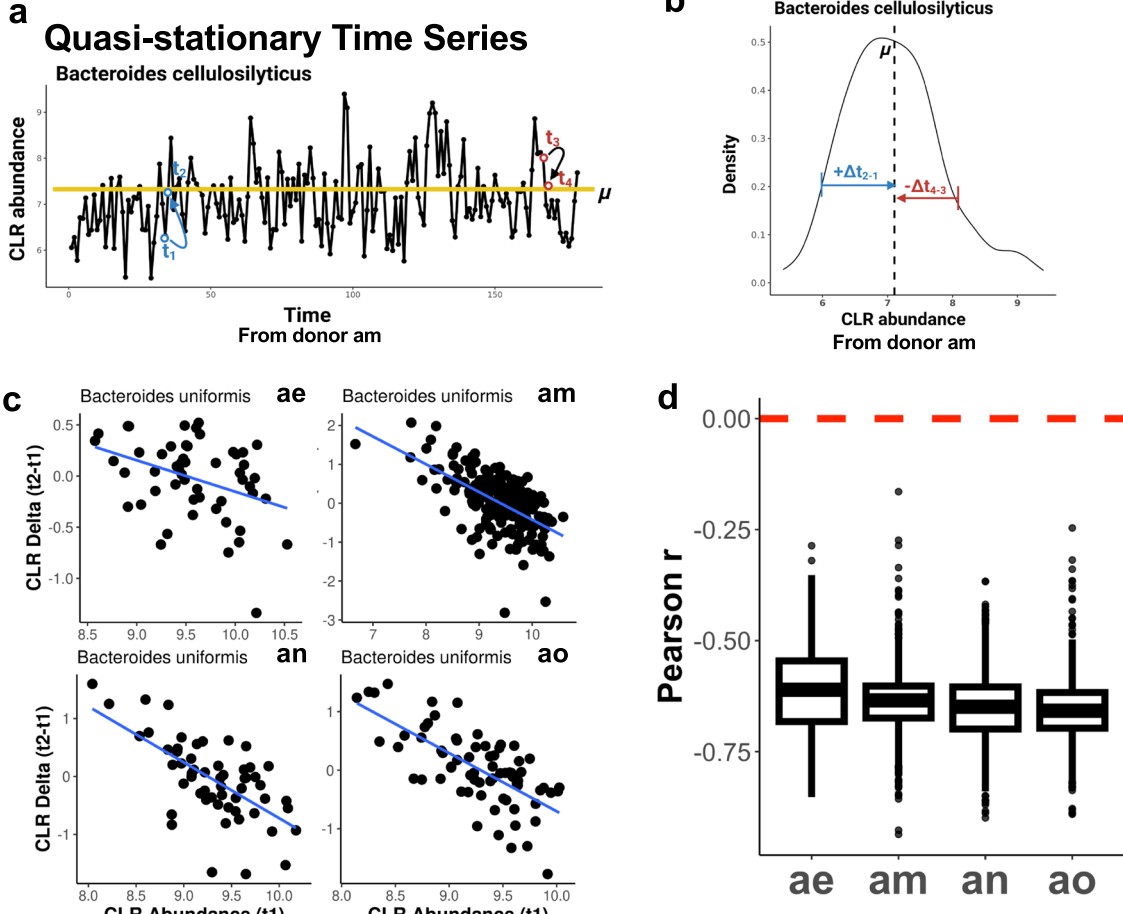

**Fig. 2 | Regression-to-the-mean effect in human microbial time series data.**
**a** Yellow line represents the mean abundance ($\mu$) of *Bacteroides cellulosilyticus* over time in donor am. Time points $t_1$ and $t_3$ indicate fluctuations below and above the mean abundance, and $t_2$ and $t_4$ show the return to the mean abundance.
**b** Distribution of time series delta values (e.g., $t_2$-$t_1$) for *Bacteroides cellulosilyticus* in donor am, which is approximately normally distributed. **c** Delta vs. abundance for *Bacteroides uniformis* time series from donors ae, am, an, and ao. **d** Box plots (showing minima, 25th percentile, median, 75th percentile, and maxima) of Pearson r values for deltas vs. abundances across all taxa time series in all four donors. Red line indicates a Pearson correlation coefficient of 0.

full complexity of the gut, such as spatial structure and specific resource usage, the LGE and its variations have been widely applied to population and community dynamics[26,27,31,32,52]. This model is defined such that the change in abundance for each taxon $i$ ($dx_i/dt$) is captured by the current abundance at time $t$, $x_i(t)$, multiplied by the maximal growth rate, $r$, and the carrying capacity ($k$) term $(1 - x_i(t)/k)$[53]. In this model, population size over time shows a sigmoidal curve, with the abundance asymptotically approaching $k$ (Fig. 4a, top panel). The derivative of this curve with respect to time yields the change in abundance, which reflects the effective rate of population growth at a given point along the curve (i.e., effective growth rate over time), and peaks during mid-log phase (Fig. 4a, middle panel). The second derivative of abundance with respect to time, which is the instantaneous change in effective growth with respect to time and is often referred to as the acceleration rate, shows a peak during the acceleration phase and a trough during the deceleration phase (Fig. 4a, bottom panel). Based on this second-derivative curve, we show the expected relationships between the effective growth rate (i.e., $dx/dt$) and abundance moving across the logistic growth curve, along the time axis (Fig. 4b). These expected relationships provide a potential path forward for inferring the in situ growth phase of a bacterial population sampled at a semi-consistent frequency from a flow-through ecosystem.

The logistic growth model is a deterministic equation. However, the observed abundances of commensal bacterial populations in the gut fluctuate due to myriad factors including interspecies competition,

resource fluctuations, technical noise, sampling noise, and stool residence time[27,32,54]. In order to approximate these fluctuations in our modeling, we introduced a stochastic term to the logistic growth model (Fig. 5a). Herein, $\sigma$ denotes the noise magnitude and $\omega(t)$ represents a white noise term. Four growth phases (i.e., acceleration, mid-log, deceleration, and stationary) were defined using the half-maximum and half-minimum, respectively, of the second derivative of the LGE curve (Fig. S4A). We simulated 100 iterations of the stochastic logistic growth equation (sLGE) for each of a range of parameterizations (see Methods), which recapitulated the expected statistical relationships between growth rates and abundances for populations consistently sampled within our four major growth phase categories (Fig. 5a–c). For example, Pearson correlations between growth rates and abundances were significantly positive in the acceleration phase and significantly negative in the deceleration phase (Fig. 5b). Mid-log phase growth was more variable, but showed little-to-no significant association between growth rates and abundances (Fig. 5b, c). These results were reproduced when combining simulation results across a wide range of parameter space and varying noise levels (Fig. S4B).

Even though we expect dietary intake to be stationary within an individual, variation in diet can drive day-to-day fluctuations in the carrying capacities of microbial populations. In order to investigate whether growth-phase specific associations between abundances and growth rates were influenced by fluctuations in carrying capacity, we added variation to $k$ in the sLGE model (Fig. S5). Fluctuations in $k$ did

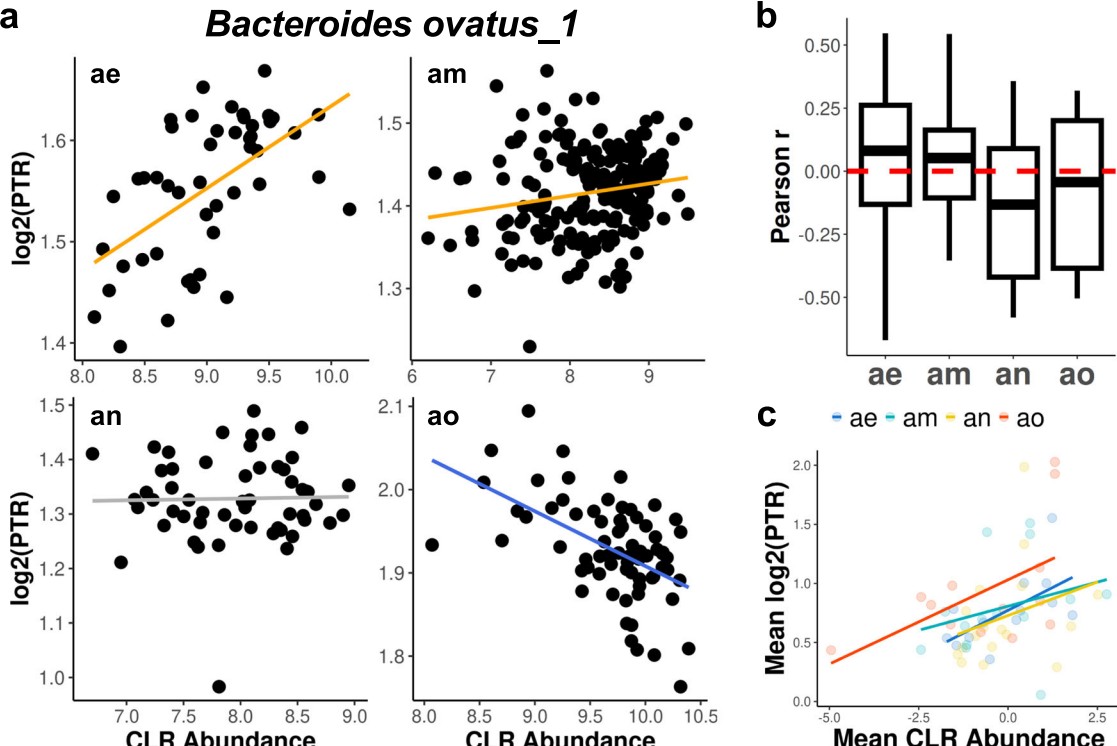

**Fig. 3 | Variable relationships between PTRs and CLR-normalized abundances across human gut microbial time series.** The ratio of sequencing coverage near the replication origin to the replication terminus for each species (i.e., peak-to-trough ratio, or PTR), was calculated using COPTR. **a** Log$_2$(PTR) and CLR-normalized abundance relationships for *Bacteroides ovatus_1* in donors ae, am, an, and ao. Orange and blue lines show significantly positive and negative linear regression coefficients (linear regression, FDR adjusted *p*-value < 0.05, respectively. Gray lines indicate no statistically significant association. Donor ae: *p* = 0.0005, donor am: *p* = 0.0317, donor an: *p* = 0.925, donor ao: *p* = 0.00005. **b** Box plots (showing minima, 25th percentile, median, 75th percentile, and maxima) of Pearson r values combined for all filtered taxa for each donor. **c** Mean log$_2$(PTR) and mean CLR-normalized abundance for all abundant taxa in each donor (*p*-values for regressions run within each donor were combined using Fisher's method; combined *p*-value = 0.005).

not alter the sigmoidal shape of the sLGE curve (Fig. S5A), and the relationships between abundances and growth rates across growth phases were preserved (Fig. S5B–C). Simulations with the noise level set at 0.2 recapitulated the correlation coefficient distributions seen in the human time series data, while maintaining the distinct growth-abundance relationships previously associated with each growth phase (Fig. S6). However, these distinct relationships could be destroyed at high enough noise levels (Fig. S6).

**Validating sLGE growth phase inferences in vitro**
To validate the relationship between growth/replication rates and abundances across growth phases, we cultured replicate *E. coli* populations in vitro and sampled them across their growth curves (Fig. 6a). *E. coli* abundances were measured as OD600 values and as the log-ratio of *E. coli* reads to phiX reads (i.e., a fixed amount of the phiX genome was spiked into each DNA extraction) from the shotgun sequencing data (Fig. 6a–c). Effective growth rates were quantified as the log$_2$(PTR) for each *E. coli* sample[55]. The relationships between the log$_2$(PTRs) and CLR-normalized *E. coli* abundances across growth phases matched the sLGE model predictions (Fig. 6b, c). Specifically, growth rates and abundances were significantly positively and negatively correlated in acceleration and deceleration phases, respectively (Fig. 6b, c). Furthermore, we saw no significant association between growth rates and abundances in mid-log and stationary phases (Fig. 6b, c). Finally, we found that samples in mid-log phase had an average log$_2$(PTR) of 1.25 ± 0.167 (± standard deviation), while samples in stationary phase had an average log$_2$(PTR) of 0.358 ± 0.059, which clearly distinguished between these phases that lacked a differentiable correlation signal.

**Inferring in situ growth phases for abundant gut commensal populations sampled in metagenomic time series**
Based on the sLGE results and in vitro validation work presented above, we assigned putative in situ growth phases to abundant gut bacterial populations from the four BIO-ML gut metagenomic time series. The average magnitude of the PTR provides additional information on whether a population is more likely to be in acceleration/ mid-log/deceleration (i.e., log$_2$(PTR) ≫ 0.358) or stationary (i.e., log$_2$(PTR) < 0.358) phase (Fig. 6). This log$_2$(PTR) threshold of 0.358 is somewhat arbitrary, based on experimental data from a single organism (Fig. 6), but it was able to classify several gut species (*Escherichia coli, Citrobacter rodentium, Lactobacillus gasseri, Enterococcus faecalis*, and several more) grown into stationary phase from prior in vitro experiments[21]. For those taxa with average log$_2$(PTRs) above the empirical stationary phase threshold, significantly positive associations (linear regression, FDR-adjusted *p* value < 0.05, with a positive beta-coefficient) between log$_2$(PTRs) and CLR abundances likely indicate acceleration phase and significantly negative associations (linear regression, FDR-adjusted *p* value < 0.05, with negative beta-coefficient) likely indicate deceleration phase. *Bacteroides cellulosilyticus, Bacteroides ovatus_1*, and *Megaspaera eldenii* showed significantly positive PTR-abundance associations within donor ae (Figs. 7A and S7). *Bacteroides xylanisolvens* had an average log$_2$(PTR) that was lower than the stationary threshold in donor am (Fig. S8). *Bacteroides ovatus_1* and *Parabacteroides distasonis* showed positive log$_2$(PTR)-CLR abundance associations, while *Alistipes finegoldii* and *Bacteroides uniformis* showed negative associations in donor am (Figs. 7a and S8). *Acidaminococcus intestini, Bacteroides xylanisolvens*, and *Odoribacter splanchnicus* showed average log$_2$(PTR) below the empirical stationary phase threshold in donor an (Fig. S9). *Alistipes shahii, Bacteroides*

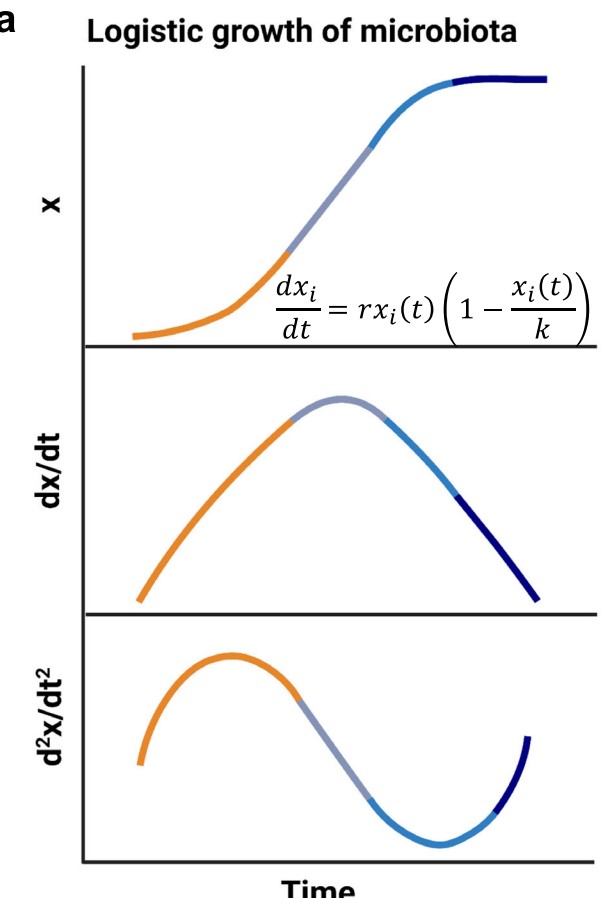

## a   Logistic growth of microbiota

$$\frac{dx_i}{dt} = rx_i(t)\left(1 - \frac{x_i(t)}{k}\right)$$

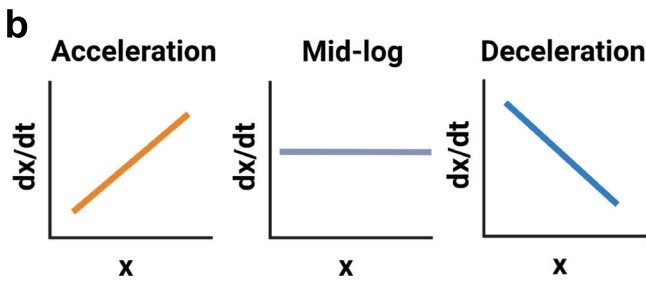

## b

**Fig. 4 | Diagram of the logistic growth equation. a** The logistic growth curve models abundance ($x$) with respect to time (top panel). Orange, gray, blue, and navy indicates acceleration, mid-log, deceleration, and stationary phases, respectively. The first derivative of the logistic growth curve shows the growth rate with respect to time (middle panel). The second derivative of the logistic growth curve shows growth acceleration with respect to time (bottom panel). **b** Expected relationships between abundance and growth rate at different locations along the logistic growth curve.

*intestinalis*, *Bacteroides thetaiotaomicron*, and *Bacteroides uniformis* showed significantly negative log$_2$(PTR)-CLR abundance associations in donor an (Figs. 7a and S9). Finally, *Favonifractor plautii* showed a positive log$_2$(PTR)-CLR abundance association and *Bacteroides fragilis*, *Bacteroides ovatus_1*, *Bacteroides uniformis*, and *Bacteroides xylanisolvens* showed negative associations in donor ao (Fig. 7a and S10). In all four donors, many taxa showed average log$_2$(PTRs) greater than the stationary threshold but without significant associations between log$_2$(PTR) and CLR abundances (Figs. 7a and S7–10). The absence of a significant association for these putatively non-stationary taxa could indicate mid-log phase, but a non-significant association could also

represent a false negative (i.e., not powered enough to detect a positive or negative association with the number of time points sampled).

We observed a slight difference in the number of significantly positive and negative PTR-abundance associations between donors ae/am, and an/ao. Donors ae and am tended to have a larger proportion of taxa in acceleration phase, while an and ao tended to have a larger proportion of taxa in deceleration or stationary phases. Interestingly, donors an and ao had a lower average defecation frequency (≤1 per day) than donors ae and am (> 1 per day). Concordantly, based on our flow-through model of the gut ecosystem (Fig. 1a), we would expect that bacterial populations would be pushed towards earlier growth phases at faster flow rates (Fig. 7b). Overall, we were able to at least partially constrain our putative phase estimates for all taxa with sufficient longitudinal data (Fig. 7a). Our approach provides a new potential path toward providing constraints on in situ growth phases for microbial populations in flow-through ecosystems.

## Discussion

Many prior studies assumed, either implicitly or explicitly, that the growth and death rates of gut bacterial populations were proportional to day-to-day changes in abundances, as measured from human stool samples. However, we outline how this assumption is likely invalid due to the fact that human gut bacterial population growth/death processes inside the intestinal tract are likely faster (minutes-to-hours) than our sampling timescales (days). Despite the fundamental mismatch between gut bacterial population dynamics and sampling timescales, we attempt to identify statistical signatures within these daily-sampled human gut time series that might provide accurate insights into in situ population dynamics.

While changes in abundance between time points do not appear to be related to population growth, metagenome-derived PTRs have been shown to be proportional to effective growth rate measures in vitro[21–23,56–58]. However, normalizing PTR values across taxa remains a challenge. For example, one might assume that the time required to replicate a particular bacterial chromosome (i.e., C-period)[21,59] would be an important conversion factor between PTRs and effective growth rates. However, prior work has shown that genome size and growth rates are uncorrelated in bacteria and archaea, and that ribosomal gene copy number explains much more variance in bacterial growth rates[60]. Ribosomal gene copy number, on average, has been shown to be higher for microbes living in the human gut, as compared to taxa in non-host-associated environments[61–64], which suggests that gut microbial taxa are optimized for higher maximal growth rates. Even if we cannot precisely convert PTRs to effective growth rates that are intercomparable across taxa, prior work has shown similar ranges of minimum and maximum PTRs across diverse gut-associated taxa in vitro[21], so we have made the coarse assumption that these values can be roughly compared across organisms (Fig. 3c).

Unlike the relationships between deltas and abundances, which were always negative (Fig. 2c, d), the relationships between PTRs and abundances were quite variable (Fig. 3a, b). While regression-to-the-mean is a plausible mechanism for the consistent negative delta-abundance relationships (Fig. 2), the underlying processes driving variable log$_2$(PTR)-abundance relationships appear to be more nuanced (Fig. 3). We turned to the sLGE to explore relationships between growth rate and abundance across different phases of growth, and we found clear diagnostic patterns (Fig. 4). Prior work has shown that the sLGE is an optimal model for capturing within-person strain- and species-level dynamics for the vast majority of human gut commensals[32]. Simulations showed how these diagnostic patterns held across a wide range of external stochastic noise levels (Fig. 5). Furthermore, fluctuations in carrying capacities, which could be interpreted as fluctuations in niche-size due to dietary variation, could not ablate these patterns (Fig. S5). However, as one might expect, adding

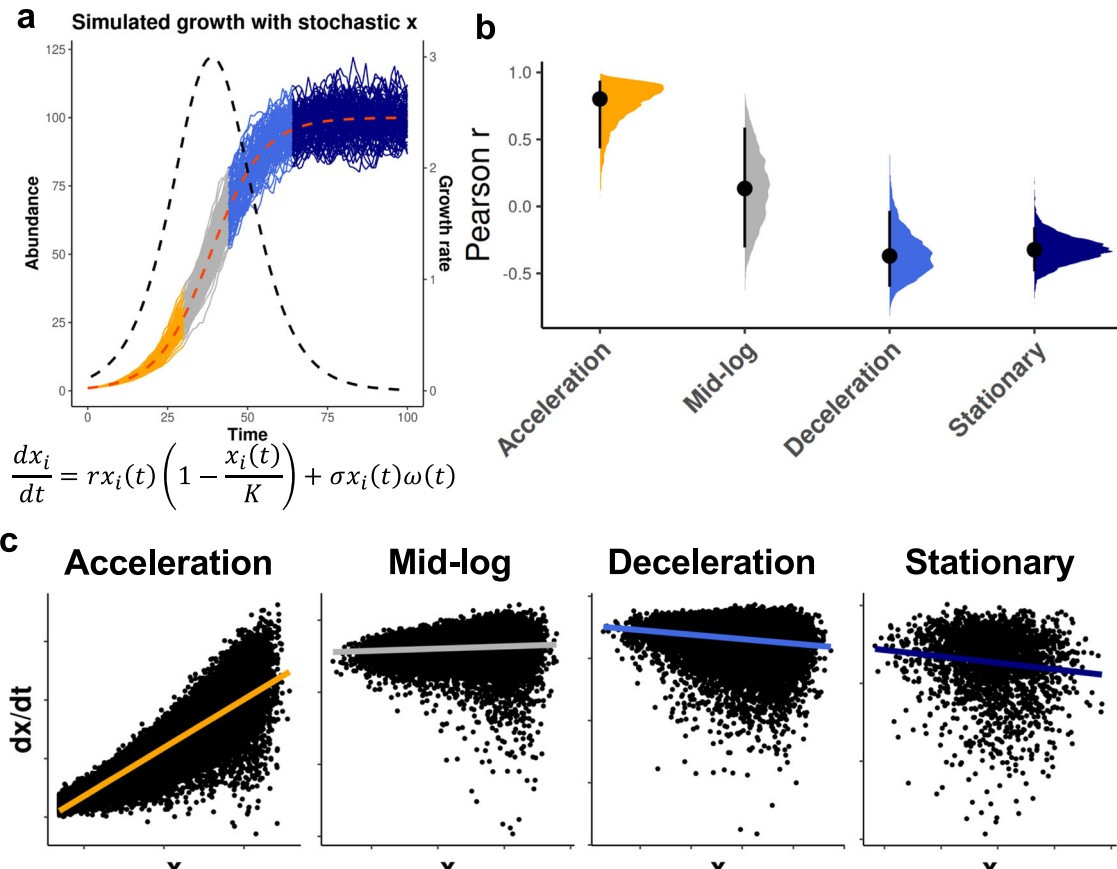

$$\frac{dx_i}{dt} = rx_i(t)\left(1 - \frac{x_i(t)}{K}\right) + \sigma x_i(t)\omega(t)$$

**Fig. 5 | Distinguishing growth phases using the stochastic logistic growth model. a** Stochastic logistic growth curves with growth rate ($r$) = 1.2, carrying capacity ($K$) = 100, and noise level ($n$) = 0.1 across 100 iterations. Major growth phase groups in orange (acceleration), gray (mid-log), blue (deceleration), and navy (stationary). **b** Pearson r values between abundances and growth rates in each of the four growth phase windows across variable model parameterizations ($r$ = 1–3, $K$ = 10-1000) and a fixed noise level ($\sigma$ = 0.1). Black circles represent the median and black bars show 95% confidence interval. **c** Scatter plots in log scale showing relationships between abundance and growth rate across the four growth phase regions defined in (**a**).

enough noise to these models eventually did override the signal (Fig. S6).

We validated the putative growth phase diagnostic patterns from our sLGE model in vitro and saw marked correspondence (Figs. 5 and 6). The in vitro data indicated that the average log₂(PTR) across the growth curve could easily distinguish between stationary phase (i.e., no effective growth) and the other phases (Fig. 6). This is particularly useful for distinguishing stationary phase from log-phase growth, which are both expected to show a null association between log₂(PTRs) and abundances (Figs. 4 and 5). We saw that *E. coli* grown in vitro showed a maximum log₂(PTR) of ~1.75 and a minimum of ~0.3, which is quite consistent with log₂(PTR) ranges observed for other organisms growing in vitro and in vivo[21]. While log₂(PTR) ranges can vary across taxa, we tentatively set a log₂(PTR) threshold of ≤0.358 to assign taxa measured in vivo to putative stationary phase, which is on the lower end of what has been observed in vitro and in vivo[21].

We applied our sLGE predictions to four human gut metagenomic time series. Consistent with our sLGE predictions, we found that individuals with higher defecation rates tended to be enriched for taxa in earlier growth phases (Fig. 7). In a recent study, we observed a similar association between PTRs and bowel movement frequency (BMF) in another independent cohort, where average community PTRs appeared to increase with increasing BMF[65]. Overall, our results reveal a promising approach to inferring in situ growth phases for abundant organisms detected in human gut metagenomic time series.

We observed that the average log₂(PTR) and average CLR abundance of a given taxon over time were positively, albeit weakly, correlated, which is consistent with exponentially-growing populations (Fig. 3c). We were also able to identify specific taxa that were abundant in stool that appeared to be in stationary phase, based on our in vitro stationarity threshold (Fig. 7a). These results are highly relevant to the metabolic modeling community. Ecological interactions within free-living and host-associated microbial communities are largely governed by exchanges of small-molecule metabolites[66,67]. Genome-scale metabolic modeling and flux-balance analysis (FBA) has been effective mechanistic tools for simulating these metabolic exchanges, especially in controlled bioreactor systems[68]. The objective function used to find a solution subspace for these bacterial FBA models is often biomass maximization, which assumes that these organisms are growing exponentially at steady state. Exponential growth is a valid assumption for organisms in acceleration or mid-log phases, and to some extent in deceleration phase, but this assumption breaks down completely in stationary phase. Prior work has demonstrated that biomass composition can change depending on the growth phase of a population, which ideally could be taken into account to more accurately model metabolic fluxes within the system[69–71]. Alternatively, organisms that are not actively growing in the distal colon could be omitted from community-scale metabolic models of colonic metabolism[72]. Overall, our work suggests that most abundant organisms in human stool are amenable to FBA, and our growth phase estimation approach allows for the identification of abundant populations that may not fit classical FBA assumptions.

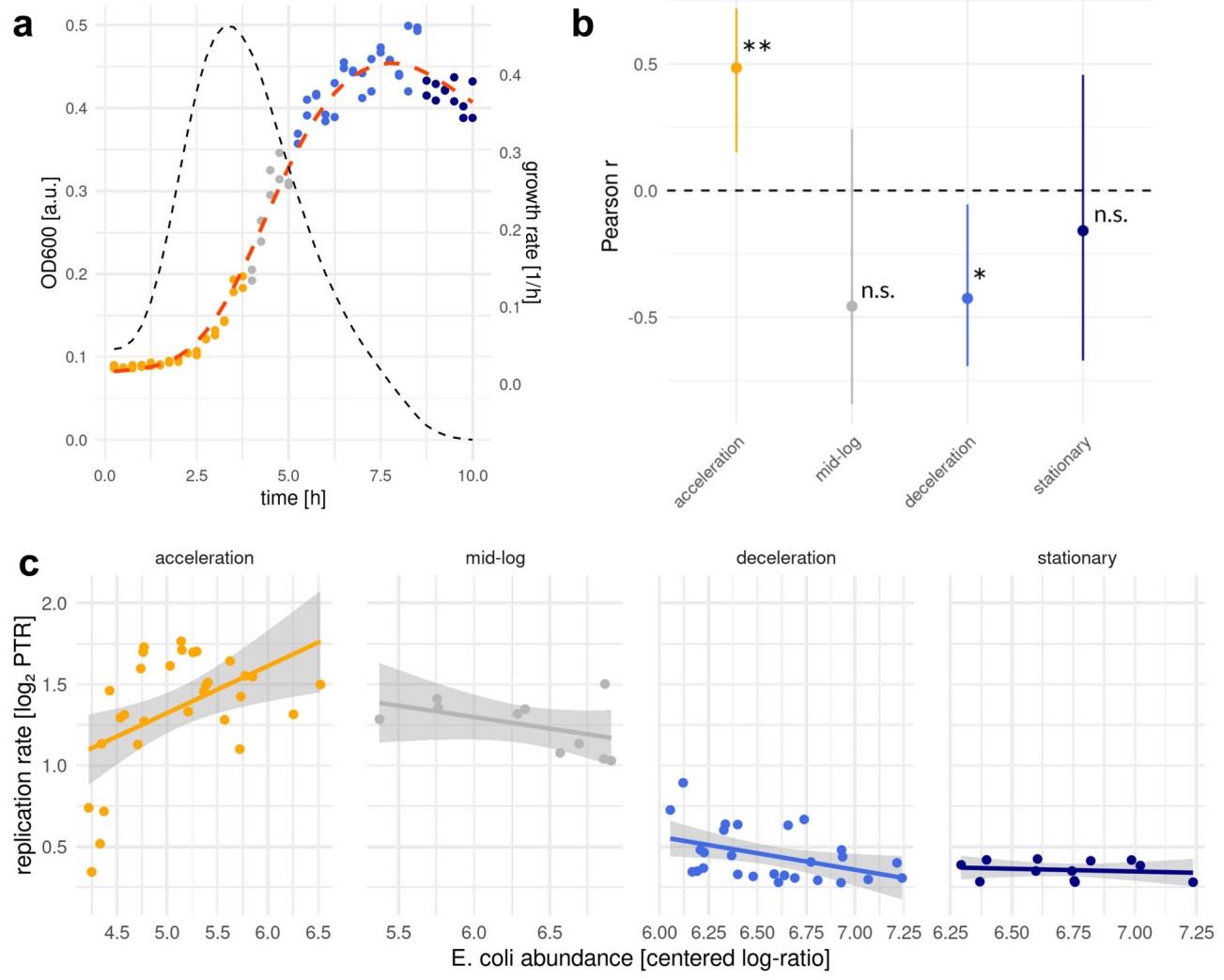

**Fig. 6 | Relationship between growth rate and abundance in major growth phases in *E.coli* populations. a** Growth curve of *E.coli* (MG1655) using OD measurements. Colors describe major growth phases. Dotted black and red lines show the growth rate derived from OD measurements and mean growth trajectory, respectively. **b** Pearson r values between abundance and growth rate in each of the four growth phase windows. Asterisks show statistical significance from two-sided correlation tests without adjustment for multiple comparisons. **: $p < 0.01$ (acceleration: $p = 0.007$), *: $p < 0.05$ (deceleration: $p = 0.003$), n.s.: not significant (mid-log: 0.184, stationary: 0.622). Black circles represent the median and black bars show 95% confidence interval. Pooled duplicate samples (4 sets of replicate cultures) for 40 time points in total were used (see Methods). **c** Scatter plots in log scale showing relationships between abundance and replication rate ($\log_2$PTR) across the four growth phase regions defined in (**a**). Gray regions represent 95% confidence intervals.

In conclusion, we provide a new path forward for the biological interpretation of metagenomic time series data generated from adult human stool samples. Our results are somewhat reassuring for cross-sectional studies, as they indicate that bacterial abundances in the gut fluctuate around stable carrying capacities within an individual, making inter-individual comparisons fairly robust. Furthermore, this suggests that multi-day averages of abundances will be even more accurate estimates of this carrying capacity, as we have suggested previously[33]. This work is especially relevant to the design and interpretation of human gut microbiome studies that aim to characterize or investigate ecosystem-scale dynamics. We hope that in situ growth phase estimation will be applied more broadly to other kinds of flow-through environments to improve our understanding of internal dynamics in these systems and provide improved constraints for mechanistic modeling of microbial communities.

**Study limitations**
One critique of our approach is that dense longitudinal sampling of human stool metagenomes is necessary for growth phase inference.

Currently, these data sets are rare, which limits the immediate use of our approach. However, interest is growing for long-term monitoring of gut microbial variation[73–75]. Through technological advances, dense metagenomic time series will become more and more common over time[76,77].

Another limitation is in our ability to relate PTRs to effective growth rates. Earlier research compared PTR curves and measured growth curves (OD600)[21], which allows for a continuous mapping between dx/dt (i.e., the effective population growth rate) and PTRs. Although our findings lack thorough quantitative estimates of in vivo effective population growth rates, we make the assumption here that in vivo PTRs are indeed related to growth. For some of our analyses, we also assume that PTRs are roughly comparable across taxa. For example, we chose a simple threshold ($\log_2$(PTR) $\leq 0.358$) for determining which taxa in the gut are in stationary phase, based on our in vitro *E. coli* experiment. However, as mentioned above, we do find that this threshold also successfully classifies many other gut species into stationary phase from prior in vitro experiments[21]. Nonetheless, future work is needed in order to develop better

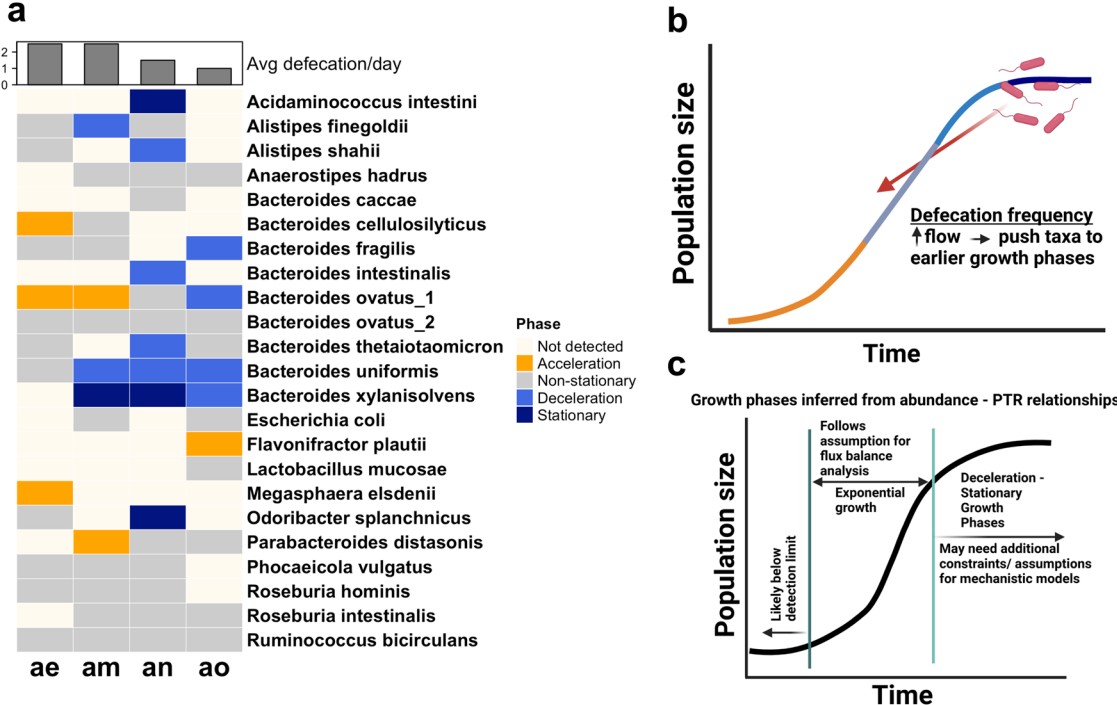

**Fig. 7 | In vivo growth phase estimation. a** We find variable relationships between $\log_2$(PTRs) and population abundances across taxa in each of the four donors, consistent with the growth phase patterns observed in sLGE simulations. Donors with higher defecation rates tended to have a larger fraction of taxa with positive $\log_2$(PTR)-abundance associations and fewer with negative associations, indicating acceleration and deceleration-stationary phases, respectively. Taxa in stationary phase were classified using an empirical threshold (average $\log_2$PTR < 0.358). Non-stationary taxa (i.e., above the stationary phase threshold, but lacking a significant correlation between $\log_2$(PTRs) and abundances) are likely in mid-log phase, but these taxa could also be in acceleration/deceleration phases (i.e., underpowered to detect the correlation). **b** We suggest that higher defecation rates (i.e., higher dilution rates) push bacterial populations towards earlier growth phases, which is consistent with our results in (**a**). **c** Growth phase estimates can be leveraged to identify taxa that are more-or-less amenable to metabolic modeling techniques, such as Flux Balance Analysis, which assumes exponential growth.

conversion factors that enable direct comparisons of PTRs across taxa.

The jury is still out on whether sLGE, in all its simplicity, is an appropriate coarse-grained model for human gut commensal dynamics. However, recent work has found that 86.6% of species time series from the same BIO-ML stool donors studied here could be optimally fit by the sLGE, using a fairly stringent significance cutoff[32]. *Phocaeicola vulgatus* was one of the few taxa that overlapped with our analysis that previously showed a poor fit to the sLGE across stool donors[32]. In our analysis *P. vulgatus* never showed a significant association between PTRs and abundances across any of the stool donors, but its average PTR was higher than the stationary phase threshold (Fig. 7). Thus, we only applied a stationary phase cutoff to this organism. We look forward to other independent research groups testing, and hopefully improving, our in situ growth phase inference approach as additional data sets become available.

## Methods
### Stationarity testing for daily nutrient intake in a human stool donor
Metadata for daily nutrient intake, excluding the time window when the donor was traveling abroad, was downloaded from David et al.[48]. We tested for stationarity in these nutrient intake time series using the augmented Dickey-Fuller (ADF) test (tseries package in R[78]), with significance threshold for stationarity at $p < 0.1$. ADF tests the null hypothesis that a unit root is present in a time series, with the alternative hypothesis being that the time series is stationary. Thus, significant $p$-values indicate stationarity of the time series. All analyses throughout the manuscript in R were conducted in R v4.2.2[79], unless stated otherwise.

### *E. coli* strain information and growth curve analysis with a microplate reader
*Escherichia coli* strain (MG1655) was streaked from a glycerol stock onto R2A agar plates (Thermo Fisher Scientific: Oxoid CM0906) and incubated overnight at 37 °C. A colony was selected using an inoculating loop and transferred to 200 mL of LB-broth (Lennox) and grown at 37 °C overnight in a shaking incubator at 225 rpm until the culture reached stationary phase. The overnight culture was then diluted in fresh LB medium to an OD of 0.51 (600 nm). The diluted culture was then chilled for ~25 minutes at ~2 °C using an ice bath to synchronize metabolic activity. The chilled culture was then aliquoted (2 μL) into a non-treated 96-well flat-bottomed plate (Thomas Scientific Cat No. 1154Q44) containing 198 μL of LB media (Lennox) in each well. The inoculated plate was then transferred to a BioTek Epoch II plate reader set to 37 °C with orbital shaking and programmed to make OD600 readings every minute for the first 60 minutes and every 5 minutes for the remainder of the experiment (~10 hours). The first set of inoculations covered plate rows A and B (n = 24), this was followed by the sequential inoculation of the next 3 sets of rows at 15-minute intervals (i.e., Set 1 = A/B: 0 min; Set 2 = C/D 15 min; Set 3 = E/F: 30 min; Set 4 = G/H 45 min). This resulted in 4 sets of replicate cultures inoculated 15 minutes apart, allowing sampling every hour for the next 10 hours, spanning 40 time points spaced 15 minutes apart. To ensure there was enough DNA for sequencing at early low OD time points (first two sample points), we pooled two wells into one sample. All samples were collected in PCR strip tubes (Axygen: PCR-0208-CP-C) and centrifuged at room temperature to pellet the cells. The supernatant was decanted and the remaining cell pellet was immediately frozen in liquid nitrogen for storage at −80 °C.

### DNA extraction, library preparation, and sequencing

Cell pellets were resuspended and transferred to 96 deep-well plates for DNA extraction using the IBI Scientific 96-well Genomic DNA Bacteria Kit (IBI Scientific: IB47295) per the manufacturer's protocol. DNA quantification was done using Qubit HS DNA assay, on a Qubit3 device. After DNA quantification, we added PhiX DNA (Thermo Fisher Scientific: SD0031) as an internal standard and run-quality monitor across all samples. A total of 500 fg PhiX DNA was added to each DNA sample before library preparation. DNA libraries were constructed following the NEBNext Ultra II FS DNA Library Prep Kit for Illumina (New England Biolabs: E7805L) and indexed using Dual Index Primer Set 2 (New England Biolabs: E7780S). Libraries were quantified again via Qubit 3, and the quality and size of libraries were checked using an Agilent Tapestation, and a D5000 high-sensitivity DNA tape assay. Libraries were pooled to 2 nM and sent to NovoGene for sequencing on a NovaSeq 6000 device (Illumina, USA). A partial lane was used for sequencing, 150 cycles, generating ~64GB (~3.3 million reads per sample) of paired-end reads.

### Shotgun metagenomics data processing and analysis

Longitudinal shotgun metagenomics sequencing data from healthy human stool samples (BIO-ML) was downloaded from NCBI BioProject accession PRJNA544527, and the associated metadata was downloaded from the associated article[33]. Raw FASTQ files from the BIO-ML cohort and from the in vitro *E. coli* experiment were filtered and trimmed using FASTP[80], removing the first 5 nucleotides of the read 5' end to avoid leftover primer and adapter sequencing not removed during demultiplexing and an adaptive sliding window filter on the 3' end of the read with a required minimum quality score of 20. Reads containing ambiguous base calls, having a mean quality score less than 20, or with a length smaller than 50nt after trimming were removed from the analysis. Taxonomic assignment on the read level was performed with Kraken2 using the Kraken2 default database[81]. Abundances on the kingdom, phylum, genus, and species ranks were then obtained using Bracken[82]. Trimmed and filtered reads were then aligned to 2,935 representative bacterial reference genomes taken from the IGG database (version 1.01) using Bowtie2[83,84]. Coverage profiles and $\log_2$ estimates of peak-to-trough ratios (PTRs) were estimated using COPTR v1.1.2 at the species level within each sample for taxa that passed our abundance threshold[55]. PTR estimates were then merged with Bracken abundance estimates, retaining only those species identified by both methods (Kraken2 and Bowtie2 alignment to IGGdb). For the in vitro *E. coli* experiment, reads were aligned to a custom database containing the *E. coli K12* strain genome (NCBI accession NC_000913.3) and the phiX174 genome (NCBI accession NC_001422.1). CLR abundances were then calculated from the read counts for the *E. coli* genome and the phiX174 genome.

The processed data containing the raw reads and $\log_2$ peak-to-trough ratios (i.e., $\log_2$(PTRs)) were read into R version 4.1.3 for analysis[79]. All plots were generated using ggplot2[85], unless indicated otherwise. Raw read counts for a given taxon within a sample were centered log-ratio (CLR) transformed[86]. Taxa that had matched $\log_2$(PTR) and CLR abundance information available across more than 5 time points within an individual, with time differences between samples less than three days, were used in subsequent analyses. Changes in normalized abundance were calculated as *Abundance changes*(*delta*) = $x(t+1) - x(t)$, where $\Delta t < 3 days$. To assess the regression-to-the-mean effect, CLR-normalized abundances were plotted against deltas for each taxon, and the regression coefficients, aggregating all microbial taxa, were plotted as boxplots (showing median and interquartile range), summarized by donor.

For each donor, to estimate the growth phase of each individual taxon, we used linear regression of CLR-normalized abundances vs. $\log_2$(PTRs), followed by a Benjamini-Hochberg *p*-value correction to control for the false discovery rate (FDR) in base R. Regression coefficients with FDR-adjusted *p*-values < 0.05 were considered significant. Taxa with average $\log_2$(PTRs) < 0.358 (experimentally-determined stationary threshold) were designated as being in stationary phase. For those taxa not designated as being in stationary phase, significantly positive or negative associations between $\log_2$(PTRs) and abundances were considered to be in acceleration or deceleration phase, respectively. Those with no correlation and an average $\log_2$(PTR) above the stationary threshold were constrained to be in mid-log phase or in acceleration/deceleration phase (i.e., if there was a false negative due to lack of statistical power in detecting a positive or negative slope). Linear regression was also used to test whether or not average CLR-normalized abundances and average $\log_2$(PTRs) were significantly associated within each donor, and *p*-values from individual tests were combined using Fisher's method[87].

### Stochastic logistic growth model simulation

The stochastic logistic growth equation (sLGE) was implemented as: $\frac{dx_i}{dt} = rx_i(t)(1 - \frac{x_i(t)}{K}) + \sigma x_i(t)\omega(t)$, where $t$ is time, $r$ is the growth rate, $x_i$ is the abundance of taxon $i$, $K$ is the carrying capacity, $\sigma$ is the noise magnitude term, and $\omega(t)$ is the noise distribution term. Using the R package sde[88], taxonomic growth was simulated with $x_{i,0} = 1$, $t_0 = 1$ to $t_{final} = 100$, for 100 iterations. The other parameters were varied as described in the results and below. To investigate the impact of noise on sLGE trajectories, noise levels were set from 0.001 to 1, with $r$ and $k$ ranging from 1 to 3 and 10 to 1000, respectively. To investigate the statistical relationships between deltas and abundances across growth phases and across model parameterizations, Pearson's R coefficients and *p*-values were calculated for each of the three growth phase categories. The growth phases for each model parameterization were defined using the non-stochastic logistic growth equation (LGE): $\frac{dx_i}{dt} = rx_i(t)(1 - \frac{x_i(t)}{K})$, the solution for which can be written as $x_i = \frac{x_{i,0}Ke^{rt}}{(K - x_{i,0}) + x_{i,0}e^{rt}}$.

The $x_i$ values for each simulated time point from solving the LGE were used to calculate the first derivative (i.e., the growth rate), which is exactly equal to the LGE. The second derivative (i.e., growth acceleration), $\frac{d^2x_i}{dt^2} = K^2 x_i(1 - \frac{x_i}{K})(1 - (\frac{2x_i}{K}))$, was calculated using solved $x_i$ values. Growth phases from the sLGE were defined using the second derivative curves. First, the intersections of the acceleration curve and the half-max, $a_1$ and $a_2$, and the half-min, $a_3$ and $a_4$, were calculated (Fig. S4A). The corresponding simulated time points of $a_j$, denoted as $s_j$, where $j = 1–4$, were then used to define growth phases as follows: lag phase: $t < s_1$; acceleration phase: $s_1 < t < s_2$; log phase: $s_2 < t < s_3$; deceleration phase: $s_3 < t < s_4$; and stationary phase: $t > s_4$. Here, lag and acceleration phases were combined, as these phases display similar delta-abundance relationships along the logistic growth curve. Conceptual diagrams were created using BioRender.

Death or dilution terms were not explicitly added to the simulated sLGE models. Here, we discuss how death or dilution rates are equivalent to changing the carrying capacity term, which has no impact on our growth phase inferences. Analytically, a decrease in abundance at a given time can be represented as a fraction of the current abundance subtracted from the LGE: $\frac{dx_i}{dt} = rx_i(t)(1 - \frac{x_i(t)}{K}) - Hx_i(t)$. Here, $H$ is the "harvest rate", which determines the proportional decrease in each timepoint in the equation. At steady state, $rx^*_i(t)(1 - \frac{x^*_i(t)}{K}) - Hx^*_i(t) = 0$, where $x^*_i(t)$ represents the fixed point. Two equilibria exist in this equation: $x^*_i(t) = 0$ and $x^*_i(t) = K(1 - \frac{H}{r})$, with the latter being asymptotically stable. As $H$ increases, the stable population size $x^*_i(t)$ decreases due to the proportional decrease in $k$. As long as $H$ does not exceed the intrinsic growth rate of gut microbes, which is expected for highly abundant and stably colonized taxa, the resulting $k$ becomes the new stable $k$. To show that variation in $k$ does not impact the relationship between growth rate and abundance, we simulated the LGE with stochastically varying $k$ by adding the stochastic term, i.e., $\sigma k_i(t)\omega(t)$, to $k_i(t)$ (Fig. S5).

In base R, simulation was performed for 100 iterations with the same noise levels ($\sigma = 0.1$) as the representative sLGE simulations with stochastic $x$. Major growth phases were defined the same way as sLGE simulations with stochastic $x$.

### Statistics and reproducibility

Data for longitudinal daily nutrient intake[48] and human stool metagenomic sequencing[33] were secondary analyses using the profiling data obtained from the original studies. Sample sizes were not calculated, experiments were not randomized, and investigators were not blinded to allocation during outcome assessments. BIO-ML donors were selected by retaining individuals with over 50 metagenomic time points, resulting in four time series (i.e., donors ae, am, an, and ao). Distinct *Bacteroides ovatus* strains across all four donors contained duplicated taxon names with unique taxonomic identifiers, and were renamed to "*Bacteroides ovatus_1*" and "*Bacteroides ovatus_2*".

### Reporting summary

Further information on research design is available in the Nature Portfolio Reporting Summary linked to this article.

## Data availability

The raw shotgun sequencing data from the in vitro experiment are publicly available from the National Center for Biotechnology (NCBI) Sequence Read Archive (SRA), BioProject accession number PRJNA942341. The raw BIO-ML metagenomic data are available under the SRA BioProject accession number PRJNA544527.

## Code availability

Nextflow pipelines implementing the processing of metagenomic shotgun sequencing data from raw reads to taxonomic abundance matrices and PTR estimates can be found at https://github.com/Gibbons-Lab/pipelines/ (metagenomics pipelines). Scripts used to analyze the data, run the sLGE simulations, and produce the figures in the manuscript have been deposited at https://github.com/Gibbons-Lab/human-microbiome-time-series-growth-phase-estimation[89].

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

## Acknowledgements

We would like to thank Shijie Zhao for suggesting that we investigate PTR-abundance relationships in the BIO-ML data set. Thanks to Pamela Troisch for help with DNA library preparation for the in vitro experiment. We would also like to thank Amy Willis, Julia Cui, and the members of the Gibbons Lab for helpful discussions of this work. SMG and CD were supported by a Washington Research Foundation Distinguished Investigator Award and by startup funds from the Institute for Systems Biology. JL was supported by the Environmental Pathology/Toxicology training grant (ES007032) and Environmental Health and Microbiome Research Center (EHMBRACE). Research reported in this publication was supported by the National Institute of Diabetes and Digestive and Kidney Diseases of the National Institutes of Health (NIH) under award number R01DK133468 (to SMG). Work conducted by JJV, JW, and NSB, as part of ENIGMA- Ecosystems and Networks Integrated with Genes and Molecular Assemblies (http://enigma.lbl.gov), a Science Focus Area Program at Lawrence Berkeley National Laboratory is based upon work supported by the U.S. Department of Energy, Office of Science, Office of Biological & Environmental Research under contract number DE-AC02-05CH11231 (to NSB).

## Author contributions

S.M.G., J.J.V. and J.L. designed and conceived the study. J.L. performed the analyzes, ran simulations, made the figures, and wrote the first draft of the manuscript. J.W. performed the in vitro experiment. CD processed metagenomic and genomic data and contributed to data analysis and figure construction. S.M.G. and N.S.B. contributed resources and mentorship. All authors contributed to writing and editing the manuscript.

## Competing interests

The authors decalre no competing interests.
