## [Peer Review File · Nature Communications]

Reviewers' comments:

Reviewer #1 (Remarks to the Author):

The manuscript reports several conflicting and rather inconclusive observations regarding the growth rate of microbes in a human gut. On one hand, authors find a negative correlation between species abundance on a given day and the mean absolute magnitude of abundance change between consecutive time points. This is interpreted as “regression-to-the-mean” growth dynamics of the logistic growth equation. On another hand, authors report inconclusive results regarding the correlation between the mean abundance of a species on a given day and the PTR measure of its growth rate. It is not clear to me why authors expected to see these two quantities to be correlated in the first place. Indeed, bacterial abundances on a given day are likely to be affected by the diet during several previous days. Changes in dietary intake will affect the carrying capacity term k rather than the growth rate term r of the logistic equation.

My main objection against the proposed work is that in their treatment of day-to-day fluctuations of microbial abundances authors ignore the effects of fluctuations in the diet. In fact, instead of simulating the stochastic logistic growth model in which the demographic stochasticity term is simply added to the equation for dx/dt , authors should have considered a model in which the carrying capacity k is randomly changing from one day to another. It is entirely possible and even plausible that the “regression-to-the-mean” dynamics shown in Fig. 2 is due to “regression-to-the-mean” dynamics of the dietary intake and not to changes in growth of resident microbes.

In addition to this major point, I also found paper to have several more minor flaws:

1) The axes in figure 2A-2C have no tick marks or units. The same is true for axes in some of the panels in Fig. 3.

2) When calculating the first and second derivatives of x_i authors do not use log-transformation of x_i which they use in all of their figures. Without log-transformation, the “growth rate” $dx/dt \sim rx$ in the exponential growth regime will also increase exponentially. This is not what the PTR method refers to as the growth rate: the PTR method measures $r(1-x/k)$ and not $rx(1-x/k)$. Hence the comparison of the model to the PTR data is flawed.

3) A minor pet peeve: authors excessively rely on abbreviations (CLR and PTR). It took me some time to realize that the term PTR or Peaks-to-Troughs-Ratio does not refer to peaks and troughs on $x(t)$ profile and instead refers to a previously published method of inferring microbial growth rates from the sequencing data. I recommend using clear and unambiguous labels at least in manuscript's main figures.

Reviewer #2 (Remarks to the Author):

Lim, Diener and Gibbons propose a new theoretical framework with which to model and interpret changes to population sizes and measured growth rates in stool samples. The key contribution is an approach for inferring the growth phase of a microbe using dense longitudinal data. The authors embark on an exciting theoretical exploration, rethinking misguided analytic choices that are prevalent in the field, such as treating changes to abundance as growth rates. The paper is well written and is mostly clear.

I do find, however, that the model the authors proposed is not substantiated enough nor is it sufficiently validated. The observations themselves appear to offer weak support to the authors' claims. Furthermore, I am not clear about the utility of the method.

Specific comments:

1. The logical argument made by the authors is that they identify a potential model (Fig. 4), driven by theoretical arguments (some qualms with these below), show that simulations under the model generate some relationship between growth rates and abundances (Fig. 5), and then go on to analyze the same relationships in time series data (Fig. 6).

However, while the distribution under the model spans the entire range of correlations between growth and abundances (-1 to 1, Fig. 5B), the correlations observed in real data are very weak ($|r| < 0.15$) and centered around 0 (Fig. 3B). In fact, the entire range of correlations observed in actual data (Fig. 3B) falls within the range observed for the log phase (Fig. 5B). This is very strong evidence *against* the model proposed by the authors.

Furthermore, the authors themselves mention that despite having dozens of densely collected samples per individual (Fig. 1B), they are unable to infer growth phase for more than half the taxa (L210-211). This cannot be explained away, and in the absence of an analysis proving otherwise, demonstrates that the observed data is not explained by the model.

Overall, the data presented by the authors supports a lack of associations between growth rates and abundances, under which one would expect to observe some distribution of weak correlations centered around 0 (Fig. 3B). This alternative hypothesis appears to be much more strongly supported than the model suggested by the authors.

2. Can the authors offer any validation for their model? Any experimental evidence? The results presented in Fig. 6A are not strong enough to support the model. Additionally, the fraction of microbes in log phase (gray) appears to be similar across all donors. Overall, the simpler, alternative

hypothesis of no correlation between growth rates and abundances appears to be better supported by the data.

3. Even if one was to accept the model proposed by the authors – is there any utility to detecting the growth phase of different bacteria? Is it related to some biological or clinical phenotype? If such densely sampled time series are required to infer growth phases for so few taxa, is this practical?

4. Why is the logistic growth equation suitable for in vivo growth? Is there any support for its application in complex communities in vivo? Importantly, there is no dilution / death terms in these equations, and therefore the model doesn't inherently model growth rates (classic, as in, divisions per unit time), but rather changes in abundance. Additionally, the abundance of a microbe cannot decline – which is obviously not true for the gut.

5. Accepting that the gut is like a chemostat / bioreactor, is there truly a meaning for the growth phases? E.g., under a steady flow of nutrients, why would bacteria enter a stationary phase? A brief review of the references listed in L65 doesn't suggest insights to this.

6. L107-109 – How is this conjecture supported? Under a model of constant dilution under steady state (i.e., 0.5-2 defecations per day), exponential growth is needed to maintain stable abundances – and would result (again) in no correlation between abundances and growth rates. There is also an underlying assumption here that growth is the only thing that affects abundances – there could be multiple other factors.

7. L14, L42-45 L224 – Can the authors provide any support for the claim regarding fast growth rates in the gut? Has anyone measured doubling times in vivo? Ref. 21/22 definitely did not, and 23 suggests much slower growth than the authors claim (and, more importantly, suggests that in vivo growth varies widely). Back-of-the-envelope calculations (Sender et al., PLoS Biology 2016) also suggest an average division time of 12-24 hours.

8. The statements regarding Figure 2A-B (L113-114) seem like a huge stretch. It seems that every abundance profile that is remotely distributed around a mean should satisfy the authors claim. What is the null model? What observed data would refute this conclusion?

9. Fig. 3C – PTR is affected by the C-period which is species-specific. Therefore, a cross species comparison does not prove anything about relationship between growth and abundances. The signal, not surprisingly, is extremely weak and noisy.

10. L52, L126, L235 – the first to show this were Korem et al. (ref 21).

11. L178-180 is misleading as this refers to simulations and not to data.

12. What is “temporally-averaged PTRs and population sizes”? (L137) this is not explained in the methods.

13. Axis labels and ticks are missing for Fig. 2a-c. I recommend that the authors also make it very clear which figures are illustrations and which are actual observations.

Below, we provide a point-by-point response to each comment and concern, along with a revised version of our manuscript. Reviewer comments are indented and shown in blue text, while our responses are in black. We believe we have thoroughly addressed all the reviewer concerns through several additional analyses, simulations, and through the generation of a new *in vitro* validation data set that strongly supports our phase-estimation method. We believe these revisions have substantially improved the quality and impact of our paper.

Reviewer #1 (Remarks to the Author):

The manuscript reports several conflicting and rather inconclusive observations regarding the growth rate of microbes in a human gut. On one hand, authors find a negative correlation between species abundance on a given day and the mean absolute magnitude of abundance change between consecutive time points. This is interpreted as “regression-to-the-mean” growth dynamics of the logistic growth equation. On another hand, authors report inconclusive results regarding the correlation between the mean abundance of a species on a given day and the PTR measure of its growth rate. It is not clear to me why authors expected to see these two quantities to be correlated in the first place. Indeed, bacterial abundances on a given day are likely to be affected by the diet during several previous days. Changes in dietary intake will affect the carrying capacity term k rather than the growth rate term r of the logistic equation.

My main objection against the proposed work is that in their treatment of day-to-day fluctuations of microbial abundances authors ignore the effects of fluctuations in the diet. In fact, instead of simulating the stochastic logistic growth model in which the demographic stochasticity term is simply added to the equation for dx/dt , authors should have considered a model in which the carrying capacity k is randomly changing from one day to another. It is entirely possible and even plausible that the “regression-to-the-mean” dynamics shown in Fig. 2 is due to “regression-to-the-mean” dynamics of the dietary intake and not to changes in growth of resident microbes.

Response: It appears we were unclear in the presentation of our results, for which we apologize. Our goal is not to identify growth rates in the gut, as this is already a solved problem (e.g., PTR estimation, rather than calculating deltas of abundances between time points). We also were not focused on demonstrating how daily abundances sampled from the human gut are steady-state population sizes that cannot be used to estimate growth (i.e., this is all part of the ‘regression-to-the-mean’ argument, which has nothing to do with the logistic growth model), which we have already described extensively in prior work ^{2,3}. These were all ancillary points that served as background to our actual goal, which was to identify the phase of growth for an organism residing within the human gut at the time it is sampled, assuming that stool represents semi-regular sampling from a steady state, flow-through ecosystem. Currently, *in situ* phase estimation in the gut is an unsolved problem in microbiome research and has major implications for metabolic modeling, which assumes exponential growth. Knowledge about *in situ* growth phases is also relevant to experimental design consideration and statistical analysis of gut

microbiome time series (i.e., averages of multiple longitudinal samples of a steady-state abundance will serve to improve estimates of an organism's carrying capacity within an individual's gut).

We provide a mathematical model (the logistic growth equation, or LGE) that provides a very specific hypothesis/justification for why we expect varying associations (i.e., positive, negative, or null) between the rate of growth and the population abundance of an organism that is sampled at different locations along its growth curve. We now also provide an *in vitro* validation of our model, which we hope further clarifies these issues (see **Fig. 6** in the revised manuscript). Finally, a recent publication from an independent lab shows how the stochastic logistic growth model is likely the most appropriate model for both species and strain-level dynamics in the human gut and that taxa in the gut tend to fluctuate around a fixed carrying capacity ⁴.

The reviewer's concern encompasses two components: a) the relationship between the regression-to-the-mean effect and the growth dynamics of the logistic growth equation; and b) the impact of diet and fluctuating carrying capacities on the model.

a) Regression-to-the-mean and logistic growth equation

In microbiome time series data, we observed that changes in a taxon's abundance from day-to-day (deltas, inferred from sequencing read counts) within a human are negatively correlated with baseline abundances (**Figs. 1-2**) ^{2,3}. One simple explanation for this pattern would be that each taxon has a stable average abundance level (i.e., a stationary, steady-state population size or carrying capacity), with some standard amount of deviation around that average across time. Randomly sampling abundances from this normal-looking distribution (e.g., see **Fig. 2B**) would give rise to a realistic-looking abundance trajectory, which would in turn necessarily yield a negative association between fluctuation size (deltas) and baseline abundance. This is what we are referring to as 'regression-to-the-mean'. An alternative explanation that fits with our LGE model would indicate that all of the taxa in these human stool time series are sampled in their deceleration phase (or, possibly, stationary phase) of growth, which is the only growth phase where we expect a strong negative association between growth rate and abundance. However, this second explanation is clearly refuted by the fact that PTRs show a diversity of positive, negative, or null associations with abundances in the same taxa that exclusively show negative associations between deltas and abundances (**Figs. 2-3**). Thus, we conclude that deltas from daily stool samples are not useful proxies for growth rate or growth phase. Overall, our regression-to-the-mean null-explanation for why deltas are always negatively associated with baseline abundances appears to be the most plausible alternative.

b) Impact of diet on logistic growth equation

Following the reviewer's suggestion, we incorporated longitudinal dietary data from David et al. (2014) into our manuscript (**Fig. S1**). Upon analyzing the densely-sampled longitudinal dietary data from Donor A, we found that all time series showed significant stationarity, according to an augmented Dickey-Fuller test ($p < 0.1$; significant p-values indicate stationarity). In addition, we

want to highlight the fact that our *in vivo* phase estimates are made in BIO-ML stool donors, which were carefully screened by the stool bank OpenBiome as potential sources of healthy stool for fecal microbiota transplants ³. These donors were subjected to a rigorous dietary, health, and lifestyle screening. They all had a stable, relatively healthy diet. These donors were continuously screened for pathogens, and they were restricted from international travel. Overall, we expect lower average variation in diet in this group of stool donors that one might expect in the general population. Finally, recent work in baboons has shown that seasonal dietary variation had a very marginal impact on variation in gut microbiome composition, suggesting that individual-specific microbiome profiles are quite stable and fairly robust to dietary fluctuations ⁵.

Nevertheless, to account for the potential impact of dietary fluctuations on the carrying capacity, we varied k directly within the sLGE (**Fig. S4**). We show that the growth-phase-specific relationships between growth rates and abundances are maintained, even with a fluctuating carrying capacity (**Fig. S4**). Basically, fluctuations in K can add additional variation in abundance levels (not in the early growth phases, but these effects become stronger in the later growth phases, as one would expect), but they do not fundamentally alter the sigmoidal shape of the LGE. This sigmoidal shape of the growth curve is the source of the varying association patterns that we see between growth rate and abundance across growth phases. We have added these results to the revised manuscript (lines 207-214).

In addition to this major point, I also found paper to have several more minor flaws:

1) The axes in figure 2A-2C have no tick marks or units. The same is true for axes in some of the panels in Fig. 3.

Response: We thank the reviewer for pointing out these oversights. Following the reviewer's comment, we have updated Figures 2A-C to include tick marks on the axes. Figure 3 already includes tick marks. We included the word "diagram" to the figure caption titles in Fig. 1 and Fig. 4 to better distinguish conceptual figures from results/data figures.

2) When calculating the first and second derivatives of x_i authors do not use log-transformation of x_i which they use in all of their figures. Without log-transformation, the "growth rate" $dx/dt \sim rx$ in the exponential growth regime will also increase exponentially. This is not what the PTR method refers to as the growth rate: the PTR method measures $r(1-x/k)$ and not $rx(1-x/k)$. Hence the comparison of the model to the PTR data is flawed.

Response: Log transformations were often used to improve the esthetics of plotting exponentially-distributed data types (e.g., abundances vs. growth rate scatter plots), but this is just for visualization and has no bearing on the outputs of the sLGE model. The peak-to-trough ratio (PTR) is a measurement of the differential sequencing coverage between the origin and terminus of replication (along a bacterium's genome), and it is used as a proxy for microbial replication/growth rates, as introduced in Korem et al. (2015) ⁶. The strong correspondence between PTRs and other measures of growth rate have been shown before ⁶. It is true that PTRs and relative abundance deltas are fundamentally different measures, but we hope that the

connections between PTRs, empirically-measured population abundances, and growth rates across growth phases have been made quite clear by our new *in vitro* validation results (Fig. 6). We made the coloring and plotting consistent with our sLGE results, for easy comparison (Figs. 5-6).

3) A minor pet peeve: authors excessively rely on abbreviations (CLR and PTR). It took me some time to realize that the term PTR or Peaks-to-Troughs-Ratio does not refer to peaks and troughs on $x(t)$ profile and instead refers to a previously published method of inferring microbial growth rates from the sequencing data. I recommend using clear and unambiguous labels at least in manuscript's main figures.

Response: We are sorry for the confusion. In the text, we state that the PTR calculation was performed as follows: "*Coverage profiles and peak-to-trough ratios (PTRs) were estimated using COPTR v1.1.2 at the species-level within each sample for taxa that passed our abundance threshold.*" However, to further minimize confusion, we have updated the Figure 3 caption with the following sentence: "*The ratio of sequencing coverage between the origin and the terminus of replication for each species (i.e., peak-to-trough ratio, or PTR), was calculated using COPTR.*"

Reviewer #2 (Remarks to the Author):

Lim, Diener and Gibbons propose a new theoretical framework with which to model and interpret changes to population sizes and measured growth rates in stool samples. The key contribution is an approach for inferring the growth phase of a microbe using dense longitudinal data. The authors embark on an exciting theoretical exploration, rethinking misguided analytic choices that are prevalent in the field, such as treating changes to abundance as growth rates. The paper is well written and is mostly clear.

I do find, however, that the model the authors proposed is not substantiated enough nor is it sufficiently validated. The observations themselves appear to offer weak support to the authors' claims. Furthermore, I am not clear about the utility of the method.

Response: We thank the reviewer for a careful reading of our manuscript. We believe that we have thoroughly addressed all of their comments and concerns, as outlined below, which has helped us to improve the quality and clarity of our work. In brief, we have included a new *in vitro* validation experiment, which beautifully matches our model predictions and strengthens our conclusions. We have also tried to better highlight the utility of this work, by underscoring how this new way of looking at longitudinal abundance fluctuations in the gut microbiome field will help prevent the application of inappropriate mechanistic models or incorrect analytical assumptions moving forward (lines 272-285 and lines 306-324). Furthermore, we provide more discussion of how identifying abundant members of the gut microbiome that are not growing exponentially will help us to build better steady-state metabolic models of the gut, which often assume exponential growth (lines 308-324).

Specific comments:

1. The logical argument made by the authors is that they identify a potential model (Fig. 4), driven by theoretical arguments (some qualms with these below), show that simulations under the model generate some relationship between growth rates and abundances (Fig. 5), and then go on to analyze the same relationships in time series data (Fig. 6).

However, while the distribution under the model spans the entire range of correlations between growth and abundances (-1 to 1, Fig. 5B), the correlations observed in real data are very weak ($|r| < 0.15$) and centered around 0 (Fig. 3B). In fact, the entire range of correlations observed in actual data (Fig. 3B) falls within the range observed for the log phase (Fig. 5B). This is very strong evidence *against* the model proposed by the authors.

Response: The distinct relationships between growth rates and abundances across growth phases hold true analytically in the LGE, and are not a consequence of stochastic processes in the sLGE model (and adding noise to the deterministic model does not ablate these relationships; **Figs. 4-5**). Thus, we are confident in the theoretical foundation of these associations. However, to address the reviewer's concerns here (and below; see the next response), we have included new data from an *in vitro* validation experiment where we sampled replicate *E. coli* populations at different points along the growth curve and generated shotgun sequencing data from these samples (**Fig. 6**). These empirical results match what we see in our simulations. We can also see in our *in vitro* results that the noise level is lower in our experiment than in our simulations, and we do not see as much uncertainty in the mid-log phase at lower noise levels (i.e., we expect high average \log_2 PTR and a null-association between \log_2 PTR and abundance; **Fig. 6**). Furthermore, we expect a wide range of associations between growth rates and abundances across taxa in the gut if these organisms are being sampled at different points along their growth curves, which seems likely. And the associations we present from four human stool donors are corrected for multiple-hypothesis testing (i.e., FDR-corrected p-values), which controls for type I error, and we provide scatter plots of these associations for the reader to assess for themselves. Finally, the empirical results indicate that we can use a lower average \log_2 PTR threshold (<0.358) to identify populations in stationary-phase.

Overall, we hope that the inclusion of the *in vitro* validation data helps to allay the reviewer's concerns about the applicability of our model predictions to *in vivo* data. And we hope that future work in this area will test out our predictions across a wider range of systems and conditions.

Furthermore, the authors themselves mention that despite having dozens of densely collected samples per individual (Fig. 1B), they are unable to infer growth phase for more than half the taxa (L210-211). This cannot be explained away, and in the absence of an analysis proving otherwise, demonstrates that the observed data is not explained by the model.

Overall, the data presented by the authors supports a lack of associations between growth rates and abundances, under which one would expect to observe some

distribution of weak correlations centered around 0 (Fig. 3B). This alternative hypothesis appears to be much more strongly supported than the model suggested by the authors.

Response: The reviewer brings up excellent points, which can only be addressed through empirical validation. Thus, as mentioned above, we have generated *in vitro* validation data for our growth phase inference approach (**Fig. 6** in the revision, and pasted below for convenience). In addition to providing more confidence in our ability to distinguish between acceleration phase and deceleration phase (i.e., positive and negative associations between growth rate and abundance), we also find that mid-log phase can be distinguished by a relatively high average PTR and a lack of association between growth rate and abundance. In addition, we are able to define an average \log_2 PTR threshold that is indicative of stationary phase (<0.358), which provides us with more power to identify growth phases *in vivo*. All together, we now are able to constrain our growth phase estimates for all measured taxa across the four donors. We have added more discussion of these points, and we have updated our *in vivo* results (Fig. 7, Figs S5-8, and lines 232-358).

Figure 6. Relationship between growth rate and abundance in major growth phases in *E. coli* populations. **A.** Growth curve of *E. coli* (MG1655) using OD measurements. Colors describe major growth phases. Dotted black and red lines show the growth rate derived from OD measurements and mean growth trajectory, respectively. **B.** Pearson r values between abundance and growth rate in each of the four growth phase windows. Asterisks show statistical significance. **: $p < 0.01$, *: $p < 0.05$, n.s.: not significant. **C.** Scatter plots in log scale showing

relationships between abundance and replication rate (\log_2 PTR) across the four growth phase regions defined in panel A.

2. Can the authors offer any validation for their model? Any experimental evidence? The results presented in Fig. 6A are not strong enough to support the model. Additionally, the fraction of microbes in log phase (gray) appears to be similar across all donors. Overall, the simpler, alternative hypothesis of no correlation between growth rates and abundances appears to be better supported by the data.

Response: See the previous response. Briefly, we now provide experimental results validating our findings (**Fig. 6**). We cultured replicate populations of *E. coli* strain MG1655 and sampled them across growth phases for shotgun sequencing. We spiked in a known quantity of phiX genome into each of the DNA extractions from these samples prior to sequencing, so that we could calculate *E. coli* biomass directly from the sequencing data. We obtained \log_2 PTR values

and CLR transformed abundance estimates for each sample. Overall, we observed a striking degree of correspondence between our model predictions and our experimental results (**Figs. 5-6**). Thus, we show both analytically and experimentally that these relationships between growth rates and abundances can be leveraged to identify a population's growth phase, which provides us with higher confidence in our growth phase estimates for abundant microbes in human gut metagenomic time series.

In addition to our validation data, a recent publication from an independent group shows how the stochastic logistic growth model is likely the most appropriate model for both species and strain-level dynamics in the human gut and that taxa in the gut tend to fluctuate around a fixed carrying capacity ⁴.

3. Even if one was to accept the model proposed by the authors – is there any utility to detecting the growth phase of different bacteria? Is it related to some biological or clinical phenotype? If such densely sampled time series are required to infer growth phases for so few taxa, is this practical?

Response: One major utility of our model is to prevent the application of inappropriate mechanistic models and improper analytical assumptions to human gut microbiome time series data. For example, there is a large number of human microbiome papers that attempt to fit Lotka-Volterra models to gut time series ^{2,7,8}, which our results suggest would be nonsensical. We also discuss how metabolic modeling assumes exponentially-growing bacterial populations. Our growth phase inference model could identify organisms that are in stationary phase in stool (e.g., taxa from the upper-gut that are no longer growing), which are no longer contributing to the metabolic activity of the microbiota in the lower colon. Based on our validation data, these stationary-phase taxa could likely be identified by having an average \log_2 PTR below some minimal threshold (<0.358), which would require fewer time points to calculate. This kind of knowledge could be leveraged to improve mechanistic modeling on colonic metabolism, using community-scale metabolic models, like those developed by our group ⁹. Additionally, our results are potentially good news for cross-sectional analyses, because they indicate that averages of longitudinal samples can provide a more accurate estimate of steady-state abundances in gut microbiomes, which may help improve signal-to-noise in microbiome-wide-association-studies. Finally, we have seen preliminary evidence that population growth rates across the microbiota are relevant to host phenotypes. For example, we have seen a positive association between bowel movement frequency and the community-averaged PTR in another independent cohort (Arivale cohort), which suggests that commensals tend to be sampled in earlier growth phases within individuals with faster bowel movement frequencies (see Fig. 2C from another preprint by our group below). This is consistent with our observation of more taxa in earlier growth phases in the two BIO-ML individuals with faster bowel movement frequencies (**Fig. 7**), and we now cite this preprint in the revision. We suspect that our results will be relevant to a wide array of clinical variables, but more work is required to explore these types of associations.

Figure 2C from Johnson et al., 2023. In an independent cohort from what is presented in this paper (i.e., the Arivale cohort), we observe higher average community-wide PTRs for individuals with higher self-reported bowel movement frequencies (BMFs), which suggests that

commensals tend to be sampled from earlier growth phases in individuals who defecate more frequently. *bioRxiv* preprint: <https://www.biorxiv.org/content/10.1101/2023.03.04.531100v1>

4. Why is the logistic growth equation suitable for in vivo growth? Is there any support for its application in complex communities in vivo? Importantly, there is no dilution / death terms in these equations, and therefore the model doesn't inherently model growth rates (classic, as in, divisions per unit time), but rather changes in abundance. Additionally, the abundance of a microbe cannot decline – which is obviously not true for the gut.

Response: Logistic growth phenomenologically describes a population that starts off small, gets exponentially bigger, and then becomes self-limited and saturates at a defined carrying capacity. These kinds of dynamics have been observed again and again in complex real-world systems and in simplified experimental set-ups. Almost without exception (maybe the reviewer can help us identify exceptions), bacteria grow by doubling (i.e., an exponential process) and they increase their biomass until they run out of resources, which is usually through a combination of self-limitation (including resource availability and quorum sensing) and competition with other taxa in the system. The specific details of this growth can vary widely, and you can get multi-phasic growth curves when organisms switch between different metabolic modes, but the phenomenological universality of sigmoidal growth curves is hard to deny. Finally, as mentioned above, we are not the only group to suggest that sLGE is optimal for modeling the dynamics of commensals in the human gut ⁴.

Additional terms can be added to the logistic growth equation, such as a death rate. However, the decrease in abundance at a given time due to a death/harvest/defecation term subtracted from the model is mathematically equivalent to lowering the carrying capacity. As the rate of harvest/death/excretion increases, the steady-state carrying capacity decreases. As long as the rate of harvest/death/excretion does not exceed the growth rate, the adjusted carrying capacity will be above zero. We show that the growth phase trends are invariant in our logistic growth model even when we add a fluctuating carrying capacity term (**Fig. S4**).

5. Accepting that the gut is like a chemostat / bioreactor, is there truly a meaning for the growth phases? E.g., under a steady flow of nutrients, why would bacteria enter a

stationary phase? A brief review of the references listed in L65 doesn't suggest insights to this.

Response: We agree that a chemostat was not the correct analogy for the human gut. To avoid confusion, we removed the diagram that depicts a chemostat from Fig. 1A. The gut is more like a flow-through bioreactor. In one end, a semi-discrete bolus of food (contains ~0 gut bacterial cells per gram) enters the system, which becomes colonized by microbes and mixed together with host substrates (e.g., bile acids and mucus) and is ultimately excreted as a semi-discrete bolus of stool (contains $\sim 10^{11}$ bacterial cells per gram). As a bolus of food travels through the gut (and as mucus is shed from the epithelium), bacterial populations must grow from low abundance (i.e., totally absent in the initial input material) to very high abundance (~ 39 trillion bacterial cells in a human colon). Overall, bacterial populations in the gut must grow fast enough on dietary and host substrates in this flow-through system to avoid dilution-to-extinction. Very little is known about what growth phases commensals are in when stool exits the colon. Bacteria that specialize on niches in the upper gastrointestinal (GI) tract, like *Bifidobacteria* or lactic acid bacteria, are likely in stationary phase by the time we sample stool, while organisms that grow more actively in the lower-colon, like many *Bacteroidetes* and *Firmicutes* species, may still be growing exponentially in stool. We have added more discussion of these points to the manuscript (lines 232-258).

6. L107-109 – How is this conjecture supported? Under a model of constant dilution under steady state (i.e., 0.5-2 defecations per day), exponential growth is needed to maintain stable abundances – and would result (again) in no correlation between abundances and growth rates. There is also an underlying assumption here that growth is the only thing that affects abundances – there could be multiple other factors.

Response: Please see our response for comments 4 and 5. Briefly, we contend that some organisms have already completed their exponential growth phase in the upper GI, while others are still growing exponentially at the time of defecation. In the first case, we expect no correlation between growth rate and abundance (possibly a weak negative correlation, as suggested in the sLGE model) and a very low average \log_2 PTR, and in the second case we expect a higher average \log_2 PTR and several potential associations depending on what part of the growth curve an organism is sampled in. All of these patterns are consistent with maintaining a steady-state carrying capacity, but the consequences for biological interpretations, mechanistic modeling, and metabolism in the colon are quite different.

7. L14, L42-45 L224 – Can the authors provide any support for the claim regarding fast growth rates in the gut? Has anyone measured doubling times in vivo? Ref. 21/22 definitely did not, and 23 suggests much slower growth than the authors claim (and, more importantly, suggests that in vivo growth varies widely). Back-of-the-envelope calculations (Sender et al., PLoS Biology 2016) also suggest an average division time of 12-24 hours.

Response: An early observation of higher average growth rates in gut-associated microbes comes from the Earth Microbiome Project paper, where they found that the average 16S copy number was significantly higher in the gut than in other kinds of host-associated and free-living environments¹⁰. Korem et al. (2015) estimates that the median doubling time in the gut is ~2.5 hours, while Brown et al. (2016) estimates much longer doubling times (~once per day)^{6,11}. It has been difficult to calibrate PTRs to doubling times, outside of those taxa that have been cultured *in vitro* (e.g., **Fig. 6**). However, as discussed above, bacterial populations in the gut experience a constant dilution rate as food is pushed through the GI tract. If they cannot grow fast enough, they will be diluted to extinction. One challenge with PTRs is that many taxa may exhibit very low log₂PTRs due to being in stationary phase. These taxa would appear to have very slow doubling times when sampled in stool, despite likely having a fast doubling time higher up in the GI tract. Therefore, we suspect that the 2.5 hr doubling time is closer to reality than the ~24 hr doubling time. We have added these points to the discussion (lines 216-228).

8. The statements regarding Figure 2A-B (L113-114) seem like a huge stretch. It seems that every abundance profile that is remotely distributed around a mean should satisfy the authors claim. What is the null model? What observed data would refute this conclusion?

Response: At this point, it is fairly uncontroversial to assert that abundant commensal bacterial populations in the human gut exhibit stable, stationary distributions over month-to-year timescales, in the absence of major perturbations or lifestyle/diet changes^{1-4,12}. We have also shown that autocorrelation in gut commensal time series tends to decay to zero within 3-5 days^{2,3}. Thus, abundance trajectories of human commensal gut bacteria, sampled every 3-5 days, that can be approximated by random sampling from a stationary distribution, which in turn naturally gives rise to this regression-to-the-mean effect that we observe across all taxa and all donors^{2,4}. This serves as our null hypothesis. Any dynamics that can be distinguished, statistically, from this random sampling process would violate this null hypothesis. For example, the patterns predicted by the LGE model violate this null hypothesis, which is exactly what we observe when we look at PTRs instead of deltas.

9. Fig. 3C – PTR is affected by the C-period which is species-specific. Therefore, a cross species comparison does not prove anything about relationship between growth and abundances. The signal, not surprisingly, is extremely weak and noisy.

Response: Korem et al. (2015) calculated PTRs for several different gut bacterial taxa grown *in vitro* (*E. coli*, *Citrobacter rodentium*, *Parabacteroides distasonis*, and *Lactobacillus gasseri*), and the ranges of observed PTRs were comparable across taxa⁶. We see something similar in the gut metagenomes from all of the 84 BIO-ML donors, where a wide range of log₂PTRs are observed across phylogenetic groups (see new **Fig. S2** in the revision, pasted below for convenience). Very low log₂PTRs (<0.358) appear to suggest stationary phase, based on our

experimental data, and this cutoff appears to conservatively identify low-replication rates across taxa as well (Fig S2, below). We see that most abundant taxa appear to be growing in the gut (\log_2 PTRs $\gg 0.358$; **Fig. S2**), with organisms in the Bacteroidia class exhibiting some of the highest \log_2 PTRs, as we would expect in the human colon. Finally, we built regression models to look at associations between \log_2 PTR and CLR abundances across taxa, using taxonomic group as a covariate. Both at the class and the species levels, we saw that the association was positive and highly significant, independent of taxonomy ($p < 0.001$).

While the reviewer states that the relationship between average \log_2 PTR and CLR abundance is extremely weak across the four stool donors (Fig. 3C), we see that the effect is consistent across individuals and that the combined p-value is quite small ($p = 0.005$). And the relationship would be even stronger if we excluded taxa that we suspect are in stationary phase (average \log_2 PTRs < 0.358). However, we think the presentation of Fig. 3C is probably fine as is, warts and all, and we hope that **Fig. S2** and the regression models that control for taxonomic groups provide additional clarification on this issue.

Figure S2. Distributions of \log_2 PTR values across 84 BIO-ML donors, broken down by phylogenetic class. We see a fairly wide range of \log_2 PTRs within each taxonomic class. The median \log_2 PTR is fairly conserved across taxonomic classes, varying between ~ 0.4 and ~ 0.7 . In a linear regression model, controlling from taxonomic group as a covariate, we see a significant positive association between \log_2 PTRs and CLR abundances at the class-level ($\beta = 0.0612$, $p = 8.359e^{-60}$). This positive taxonomy-controlled association is preserved at the species-level ($\beta = 0.0101$, $p = 0.0006$).

10. L52, L126, L235 – the first to show this were Korem et al. (ref 21).

Response: Thank you. We have updated the references accordingly.

11. L178-180 is misleading as this refers to simulations and not to data.

Response: We now make sure to point out what data are derived from simulations and what data are derived from data. We have now made extensive edits throughout the manuscript to incorporate our *in vitro* validation data, which provides additional empirical support for our conclusions beyond our simulations.

12. What is “temporally-averaged PTRs and population sizes”? (L137) this is not explained in the methods.

Response: PTRs for each taxon sampled over time within an individual were averaged and these averages were then correlated with time-averaged within-person CLR abundances the same taxa. We have added additional clarification to the manuscript (lines 157-159).

13. Axis labels and ticks are missing for Fig. 2a-c. I recommend that the authors also make it very clear which figures are illustrations and which are actual observations.

Response: We have updated Figures 2A-C to include tick marks in the axes. Figure 3 already includes tick marks. We included the word “diagram” to the figure caption titles in Fig. 1 and in Fig. 4 to further distinguish data/results figures from conceptual/cartoon figures.

Citations

1. David, L. A. *et al.* Host lifestyle affects human microbiota on daily timescales. *Genome Biol.* **15**, R89 (2014).
2. Gibbons, S. M., Kearney, S. M., Smillie, C. S. & Alm, E. J. Two dynamic regimes in the human gut microbiome. *PLoS Comput. Biol.* **13**, e1005364 (2017).
3. Poyet, M. *et al.* A library of human gut bacterial isolates paired with longitudinal multiomics data enables mechanistic microbiome research. *Nat. Med.* **25**, 1442–1452 (2019).
4. Wolff, R., Shoemaker, W. & Garud, N. Ecological Stability Emerges at the Level of Strains in the Human Gut Microbiome. *mSystems* (2023) doi:10.1128/mbio.02502-22.
5. Björk, J. R. *et al.* Synchrony and idiosyncrasy in the gut microbiome of wild baboons. *Nat*

- Ecol Evol* **6**, 955–964 (2022).
6. Korem, T. *et al.* Growth dynamics of gut microbiota in health and disease inferred from single metagenomic samples. *Science* **349**, 1101–1106 (2015).
 7. Fisher, C. K. & Mehta, P. Identifying keystone species in the human gut microbiome from metagenomic timeseries using sparse linear regression. *PLoS One* **9**, e102451 (2014).
 8. Joseph, T. A., Shenhav, L., Xavier, J. B., Halperin, E. & Pe'er, I. Compositional Lotka-Volterra describes microbial dynamics in the simplex. *PLoS Comput. Biol.* **16**, e1007917 (2020).
 9. Diener, C., Gibbons, S. M. & Resendis-Antonio, O. MICOM: Metagenome-Scale Modeling To Infer Metabolic Interactions in the Gut Microbiota. *mSystems* **5**, (2020).
 10. Thompson, L. R. *et al.* A communal catalogue reveals Earth's multiscale microbial diversity. *Nature* **551**, 457–463 (2017).
 11. Brown, C. T., Olm, M. R., Thomas, B. C. & Banfield, J. F. Measurement of bacterial replication rates in microbial communities. *Nat. Biotechnol.* **34**, 1256–1263 (2016).
 12. Caporaso, J. G. *et al.* Moving pictures of the human microbiome. *Genome Biol.* **12**, R50 (2011).

REVIEWER COMMENTS

Reviewer #2 (Remarks to the Author):

In this revised version the most major change presented by the authors is the addition of experimental data from monocultures of *E. coli*, as well as some additional theoretical arguments and simulations. While the paper is improved, I find that my major concerns have been somewhat side stepped and not adequately addressed. My bottom line is while the additional validation is an improvement to the paper, my concerns regarding the adequacy of the model to explain the observed data and the utility of the method stand.

Below I explain my remaining concerns in light of the authors' responses as well as add additional comments regarding new analyses. Numbers correspond to my original comments.

1. The authors still offer no explanation to the very substantial difference between the strong correlations observed in simulations under the model (~ -0.7 to 1 in Fig. 5B and S4 and -1 to 1 in Fig. S3) and the very weak correlations observed in actual data (-0.15 to 0.15 , Fig. 3B). The authors did not offer a substantive explanation to this difference: the fact that adding noise to the deterministic model doesn't ablate the sLGE correlations indicate that increased noise in empirical data is unlikely the result; the confidence of the authors has no bearing on this discrepancy; and the in-vitro data also shows much stronger associations (~ -0.6 to ~ 0.7). These in vitro results indeed match the simulations – but neither the in vitro results nor the simulations match data from actual humans.

I find the revision to figure 6, with the addition of “non-stationary”, to be not very compelling. This doesn't substantiate the model, but is rather simply a translation of some empirical threshold observed in an in vitro experiment to in vivo data.

3. This comment was only partially addressed. The results from the arrivale cohort discuss average PTR and not associations with abundances. The impractical need for very long time-series has not been addressed at all.

4. My comment was not about the phenomenological suitability of sLGE to model growth curves. Claiming otherwise, as the authors do, is a strawman argument.

I instead pointed out that dx/dt in sLGE does not correspond to growth rates (divisions per unit time) but rather to change in abundance that encompasses both growth and death. The derivation showing equivalence to changing the carrying capacity (which in any case does not appear to speak

to the heart of my argument) is made under $dx/dt=0$, which is not applicable to the entirety of the model.

6. (Now L116-118) I still find very weak support for this statement. My point is that in general, bacteria will need to grow exponentially (somewhere in the GI tract) or otherwise they would be eliminated. This would not cause a positive correlation between abundances and growth rates. In general, the model offered in Fig. 1A is very simplistic, ignoring niches, mucosal adhesion, and the general complexity of the GI tract. This is fine – all models are wrong. But the conclusions from this model need to be useful.

7. I did not understand the argument regarding the earth microbiome project. To my knowledge growth rates in the gut microbiome have never been measured. As I mentioned in my original comment, Korem et al. and Brown et al. both measure PTR, which is a unitless measure that CANNOT be translated to growth rates without knowing the C-period. Gibson et al., as I mentioned, suggests slow and variable growth rates. The constant dilution rate mentioned by the authors implies a division time of 12-24 hours, as I mentioned in my original comment, and not fast growth.

The authors are free to make their own guesses and arguments regarding in vivo growth rates, but they have to be presented as such without citations to papers that either did not measure growth rates or which claim otherwise. The current phrasing across the manuscript might lead to the inadvertent effect of having an unproven conjecture being treated as a fact in the field (see for example the 10:1 bacterial:human cells ratio).

9. As mentioned above, PTRs cannot be easily translated to growth rates and a cross-species comparison is non-sensical even if the range is similar (a claim which Fig. S2, showing distinctly different distributions, does not seem to support). The taxonomy-corrected associations presented at L171-172 are highly powered – statistical significance is not as meaningful as the strength of the association (effect size). Some symbols were not properly rendered, but this seems like a very weak correlation. The best way to address this is show correlations within species – which the authors do not do for a reason that eludes me.

New comments:

14. As for the paper from the Garud group, it actually showed that sLGE only holds only in ~70-80% of species examined within the same dataset. What is the correspondence between species where sLGR doesn't hold according to the results from the Garud paper and species in which a significant association between growth and abundances is detected?

15. The in vitro experiment (Fig. 6A) shows very slow growth rates compared to what is usually known for *E. coli* in vitro (~30 minutes doubling time). Why is that the case?

16. Given the concerns regarding the interpretation of PTR as growth rate, setting a single threshold based on a single experiment in a single microbe (0.358) seems to be a questionable choice (especially as the threshold chosen is the mean of the stationary phase and not the max). Perhaps some fraction of mean for the microbe is better suited.

Reviewer #3 (Remarks to the Author):

The authors have more than adequately responded to the concerns raised by Reviewer 1, who unfortunately was not available to directly review their response. Reviewer 1's major concern was the applicability of the sLGE, as diet could be variable over time, leading to changes in the carrying capacity K . The authors responded by:

- 1) Showing their modeling results were robust to changes in K
- 2) Showing that diet tended to be stationary over time in a separate study
- 3) Highlighting that the hosts chosen for the BIO-ML study may reasonably be expected to have steady diets, given the careful screening/monitoring these hosts were subject to
- 4) Referencing other recent longitudinal studies which have shown the adequacy of the sLGE in describing commensal gut microbe time series.

Overall, I am confident that the sLGE is not only an appropriate model for describing these abundance trajectories, but also is useful in estimating the growth phase of these populations.

Other minor points regarding figures and abbreviations were also adequately addressed.

No further response to the points raised by Reviewer 1 are necessary, in my view. I recommend the manuscript be accepted for publication.

Reviewer #2 (Remarks to the Author):

In this revised version the most major change presented by the authors is the addition of experimental data from monocultures of *E. coli*, as well as some additional theoretical arguments and simulations. While the paper is improved, I find that my major concerns have been somewhat side stepped and not adequately addressed. My bottom line is while the additional validation is an improvement to the paper, my concerns regarding the adequacy of the model to explain the observed data and the utility of the method stand.

Response: We thank the reviewer for their careful consideration of our work. We did our best to understand the prior concerns, and to address them as fully as we were able with new experiments, data sets, and additional analyses. We hope that the following revisions and analyses help convince the reviewer of the validity and utility of our study. We have also included a 'Study limitations' section at the end of the manuscript, so that we are transparent about the limitations of our model in terms of both utility and validity.

Below I explain my remaining concerns in light of the authors' responses as well as add additional comments regarding new analyses. Numbers correspond to my original comments.

1. The authors still offer no explanation to the very substantial difference between the strong correlations observed in simulations under the model (~ -0.7 to 1 in Fig. 5B and S4 and -1 to 1 in Fig. S3) and the very weak correlations observed in actual data (-0.15 to 0.15 , Fig. 3B). The authors did not offer a substantive explanation to this difference: the fact that adding noise to the deterministic model doesn't ablate the sLGE correlations indicate that increased noise in empirical data is unlikely the result; the confidence of the authors has no bearing on this discrepancy; and the *in-vitro* data also shows much stronger associations (~ -0.6 to ~ 0.7). These *in vitro* results indeed match the simulations – but neither the *in vitro* results nor the simulations match data from actual humans.

I find the revision to figure 6, with the addition of “non-stationary”, to be not very compelling. This doesn't substantiate the model, but is rather simply a translation of some empirical threshold observed in an *in vitro* experiment to *in vivo* data.

Response: We believe this is a simple misunderstanding that we failed to clarify in the last round of review. The previous version of Fig. 3B was plotted with linear regression coefficients, which are expected to be smaller than Pearson correlation coefficients as the range in CLR abundance is over a magnitude higher than that of $\log_2(\text{PTR})$. We have replotted Fig. 2D and Fig. 3B using Pearson r values instead of regression coefficients and now clearly show a comparable correlation coefficient distribution range for our *in vivo* data (-0.5 to 0.5) as we see in our simulated and *in vitro* results. Sorry for any confusion there.

As we mentioned in the text, if we were to add enough noise to the system then we could destroy any signal. We have added additional simulations to show that parameterizations exist where model correlation coefficients match empirical coefficients and that enough noise will destroy our ability to estimate growth phase (lines 226-229; Fig. S6). The main point we are

trying to make is that logistic growth gives rise to the varying directionality of associations between x and dx/dt along the growth curve, which we think is quite robust.

We agree that our empirical threshold for non-growing, stationary populations is imperfect due to the reliance on *in vitro* data from a single taxon. We now mention this point in our limitations section (lines 387-394). However, the results presented in Korem et al. shows that our threshold is applicable to many gut commensal taxa grown *in vitro*, not just *E. coli*¹. We are likely being overly conservative in classifying stationary phase taxa (i.e., false negatives appear to be more likely than false positives, based on prior *in vitro* data)¹. We have included further discussion of these points below and in the revised manuscript (lines 253-257 and 328-338).

3. This comment was only partially addressed. The results from the arrivale cohort discuss average PTR and not associations with abundances. The impractical need for very long time-series has not been addressed at all.

Response: The reviewer is correct that it may currently be impractical to obtain long, dense time series from many human donors at a mass scale. We have included this point in our limitations section (lines 379-383). However, there are emerging technologies (e.g., smart toilets^{2,3}) that enable easier collection of these types of data sets and the costs of data generation continue to fall year-to-year. We anticipate that dense gut metagenomic time series will become increasingly available over time. Beyond the conceptual advances of our work, which have intrinsic value beyond any practical applications, we believe that our methods will become more and more applicable to real-world data in the future. We have added these points to the discussion (lines 379-383).

4. My comment was not about the phenomenological suitability of sLGE to model growth curves. Claiming otherwise, as the authors do, is a strawman argument. I instead pointed out that dx/dt in sLGE does not correspond to growth rates (divisions per unit time) but rather to change in abundance that encompasses both growth and death. The derivation showing equivalence to changing the carrying capacity (which in any case does not appear to speak to the heart of my argument) is made under $dx/dt=0$, which is not applicable to the entirety of the model.

Response: We apologize if we misunderstood the reviewer's point initially. We completely agree and understand that there is a difference between the intrinsic growth rate (the ' r ' term in the model) and the change in abundance at a given time point (dx/dt). Here, dx/dt reflects the effective rate of population growth at a given point along the curve, which is indeed a balance between immigration/growth and emmigration/death processes. Varying r , k , or death/dilution in the model does not influence the changing relationships between x and dx/dt as you move along the growth curve, with the exception where the death/emmigration rate overcomes growth and collapses the curve completely. These relationships arise due to the sigmoidal geometry of the curve, which is warped, but not destroyed, by variation in these parameters, except under

the edge case representing population collapse that is not relevant to our system (i.e., observed taxa in the gut are able to maintain their steady state population sizes – otherwise they would not be observed). This result holds at any point along the curve, not just where $dx/dt=0$.

6. (Now L116-118) I still find very weak support for this statement. My point is that in general, bacteria will need to grow exponentially (somewhere in the GI tract) or otherwise they would be eliminated. This would not cause a positive correlation between abundances and growth rates. In general, the model offered in Fig. 1A is very simplistic, ignoring niches, mucosal adhesion, and the general complexity of the GI tract. This is fine – all models are wrong. But the conclusions from this model need to be useful.

Response: We agree that our model is very simple, and ignores the complexities of spatial structure in the gut. Nonetheless, we think that this model offers a parsimonious explanation for the varying relationships observed between PTRs, which we see as equivalent to the effective growth rate at a given point on the logistic curve (i.e., dx/dt), and abundances. As mentioned above, we are referring to the effective growth rate at a given point on the growth curve (i.e., dx/dt), not the intrinsic growth rate of the organism (i.e., r). We do indeed expect, cross-sectionally, to see an average positive association between effective growth and population abundance if an organism is at or near its exponential growth phase at the time of sampling. The reviewer is correct that we would not expect the same relationship between the growth rate (intrinsic or effective) and abundance if organisms were in stationary phase for a substantial period of time (i.e., we would expect effective growth to be zero, and the relationship between intrinsic growth, if it could be measured, and abundance would be warped by other ongoing stationary phase forces, like dilution or death). Indeed, we think this further supports our contention that these organisms tend to still be actively growing as we sample them in stool. As mentioned above, we have included a new 'Study limitations' section in the discussion that better summarizes the caveats and limitations of our model (lines 379-404). Overall, we would love for other researchers to expand upon our work, and perhaps identify better models, but we are putting forward this rather simple model as a starting point.

7. I did not understand the argument regarding the earth microbiome project. To my knowledge growth rates in the gut microbiome have never been measured. As I mentioned in my original comment, Korem et al. and Brown et al. both measure PTR, which is a unitless measure that CANNOT be translated to growth rates without knowing the C-period. Gibson et al., as I mentioned, suggests slow and variable growth rates. The constant dilution rate mentioned by the authors implies a division time of 12-24 hours, as I mentioned in my original comment, and not fast growth.

The authors are free to make their own guesses and arguments regarding in vivo growth rates, but they have to be presented as such without citations to papers that either did not measure growth rates or which claim otherwise. The current phrasing across the manuscript might lead to the inadvertent effect of having an unproven conjecture being treated as a fact in the field (see for example the 10:1 bacterial:human cells ratio).

Response: The Earth Microbiome Project (EMP) example was meant to make the (established) link between 16S gene copy number and growth rate ⁴⁻⁶. In general, the EMP paper showed that bacteria living in the human gut tend to have a higher 16S copy number, on average, than organisms living in non-host environments, suggesting that gut microbes may be optimized for higher maximal growth rates than other environmentally-associated microbes ⁷.

We agree that we cannot provide an exact quantitative estimate of *in vivo* growth rates that can be compared across taxa based on PTRs, and we have now stated this clearly in the text (lines 301-314). We expect the C-Period to have a weak influence over the effective growth rate, compared to other parameters. For example, prior work has shown that genome size and growth rate are essentially uncorrelated in bacteria and archaea ⁸. Ribosomal gene copy number explains much more variance in bacterial growth rates ⁸, which as we show above, is also higher in the animal gut compared to other environments ⁷. PTRs can tell us whether or not an organism is actively growing, with some magnitude information about the relative rate of effective growth for that particular taxon. PTRs may be roughly comparable across taxa, based on similar minimum and maximum PTRs observed across taxa grown *in vitro* ¹. Prior papers have looked at PTRs of the same organism sampled *in vivo* and *in vitro*, which often shows higher maximal PTRs achievable *in vitro* and a similar lower PTR threshold for stationary phase populations *in vivo* and *in vitro* (e.g., Fig. 3A in Korem et al., 2015) ¹. Korem et al. also measured growth curves (OD600) along with PTR curves, similar to what we present in our revised manuscript, which allows for a continuous mapping between dx/dt and PTRs. The fact that this continuous mapping can be made is all that matters for the x vs. dx/dt associations that we use to identify putative growth phases. However, we need to be cautious when comparing PTRs across taxa, which we have reiterated in our 'Study limitations' section. We have added discussion of these points, along with the relevant references, to the revised manuscript (lines 384-394).

Ultimately, the only claims we need to make about *in vivo* growth for the purposes of this study is that dx/dt and PTRs are monotonically related within a species and that microbes need to grow fast enough to keep up with mortality and dilution rates in order to maintain their observed stable population sizes ⁹. See the lines referenced above, where we clarify these points for the reader.

9. As mentioned above, PTRs cannot be easily translated to growth rates and a cross-species comparison is non-sensical even if the range is similar (a claim which Fig. S2, showing distinctly different distributions, does not seem to support). The taxonomy-corrected associations presented at L171-172 are highly powered – statistical significance is not as meaningful as the strength of the association (effect size). Some symbols were not properly rendered, but this seems like a very weak correlation. The best way to address this is show correlations within species – which the authors do not do for a reason that eludes me.

Response: PTR distributions overlap greatly across taxa, as we see in Fig. S2 of our revision, or in Fig. 4 of Korem et al. However, the reviewer is correct that we have no quantitative basis for mapping PTR values across taxonomic groups, apart from prior *in vitro* data ¹. We tried to be

conservative in our phase definitions and in our cutoffs, to avoid this issue, as discussed above. Regardless, we still see significant positive cross-sectional associations between abundances and PTRs, even when controlling for taxonomy at multiple taxonomic levels. We recognize that these associations are rather noisy, which is not terribly surprising given the many possible sources of biological variability across stool donors, but these associations are nonetheless statistically significant. As suggested, we have now included an additional figure, showing several examples of abundance-PTR associations for prevalent bacterial species across the BIO-ML stool donors (Fig. S3). Although not statistically significant on a taxon-by-taxon basis, we observed a bias towards positive regression slopes for numerous taxa detected in the gut cross sectionally across all 84 donors. As we report in the text, when combined in a single regression model, with taxonomic ID as a covariate, this average positive association rises to the level of significance. For three species, we actually saw negative associations: *Alistipes shaii*, *Alistipes finegoldii*, and *Odoribacter splanchnicus* (FDR-adjusted $p < 0.1$). These taxa also tended to show negative associations within individuals over time (Fig. 7). Prior work by our team has found that gut commensal taxa tend to have conserved abundance levels (or carrying capacities) across humans^{9,10}. The negative associations in Fig. S3 hints that, for some taxa, a similar steady state may be present across people and that information about growth phase may be embedded in cross-sectional data as well, albeit noisier due to inter-individual variation in diet, behavior, and stool transit time.

New comments:

14. As for the paper from the Garud group, it actually showed that sLGE only holds only in ~70-80% of species examined within the same dataset. What is the correspondence between species where sLGR doesn't hold according to the results from the Garud paper and species in which a significant association between growth and abundances is detected?

Response: We believe that the applicability of sLGE to 86.6% of species trajectories (this is the fraction reported in that paper), using a conservative p-value cutoff, in the same BIO-ML data set is a great justification for our overall approach¹¹. There was a small subset of species identified in the Garud group paper, where the sLGE model did not appear to be optimal. The species that most often came up as a sub-optimal fit to sLGE across donors in the Wolff et al. paper was *Phocaeicola vulgatus* (formerly *Bacteroides vulgatus*). In our analysis, we labeled *Phocaeicola vulgatus* as 'non-stationary', as it did not show a significant association between its abundances and PTRs and its average PTR was well above our stationarity threshold (Fig. 7). Given that only one of the taxa that consistently showed a sub-optimal fit to sLGE overlapped with our study (*P. vulgatus*), and we did not constrain its growth phase beyond stating that it is not in stationary phase (based on its high PTR), we do not think this impacts our results or conclusions. We have added these points to the discussion (lines 395-404).

15. The in vitro experiment (Fig. 6A) shows very slow growth rates compared to what is usually known for *E. coli* in vitro (~30 minutes doubling time). Why is that the case?

Response: The low growth rates were due to an incorrect calculation when adding the second y-axis – thank you for catching that. We have now corrected the axis. The highest observed

growth rate in the experiment was $\sim 0.45 \text{ h}^{-1}$ corresponding to a doubling time of $\sim 1.5 \text{ h}$. This is slower than the optimal growth rate of *E. coli* mentioned by the reviewer. One potential reason for this lower observed growth rate was that cultures were grown up overnight and then kept at $\sim 2^\circ\text{C}$ for several minutes prior to transferring them to the 96-well plates, over staggered time intervals, to keep them synchronized in stationary phase prior to the start of the experiment. The observed growth rate is in line with what has been reported previously for *E. coli* cultures grown from lower temperatures (e.g., Fig. 1A, Ferrer et al. ¹²).

16. Given the concerns regarding the interpretation of PTR as growth rate, setting a single threshold based on a single experiment in a single microbe (0.358) seems to be a questionable choice (especially as the threshold chosen is the mean of the stationary phase and not the max). Perhaps some fraction of mean for the microbe is better suited.

Response: We agree with the reviewer that this is an imperfect method for choosing a threshold. However, we do find that this threshold also successfully classifies many gut species (*Escherichia coli*, *Citrobacter rodentium*, *Lactobacillus gasseri*, *Enterococcus faecalis*, and several more) into stationary phase from prior *in vitro* experiments ¹. Indeed, our $\log_2(\text{PTR})$ threshold is a bit conservative, as other taxa observed to be in stationary phase exhibit PTRs between 1.2 and 1.3, or $\log_2(\text{PTR})$ s of 0.26 - 0.37. Thus, we are inclined to keep this threshold in the current manuscript, with some additional discussion of the caveats. We have added these points to the discussion (lines 328-338 and 383-394).

Citations

1. Korem, T. *et al.* Growth dynamics of gut microbiota in health and disease inferred from single metagenomic samples. *Science* **349**, 1101–1106 (2015).
2. BiomeSense Closes Oversubscribed \$3 Million Funding Round.
<https://www.businesswire.com/news/home/20230518005056/en/BiomeSense-Closes-Oversubscribed-3-Million-Funding-Round> (2023).
3. Ge, T. J. *et al.* Passive monitoring by smart toilets for precision health. *Sci. Transl. Med.* **15**, eabk3489 (2023).
4. Klappenbach, J. A., Dunbar, J. M. & Schmidt, T. M. rRNA operon copy number reflects ecological strategies of bacteria. *Appl. Environ. Microbiol.* **66**, 1328–1333 (2000).
5. Lin, Q., De Vrieze, J., Fang, X., Li, L. & Li, X. Microbial life strategy with high rRNA operon copy number facilitates the energy and nutrient flux in anaerobic digestion. *Water Res.* **226**, 119307 (2022).

6. Li, J. *et al.* Predictive genomic traits for bacterial growth in culture versus actual growth in soil. *ISME J.* **13**, 2162–2172 (2019).
7. Thompson, L. R. *et al.* A communal catalogue reveals Earth’s multiscale microbial diversity. *Nature* **551**, 457–463 (2017).
8. Westoby, M. *et al.* Cell size, genome size, and maximum growth rate are near-independent dimensions of ecological variation across bacteria and archaea. *Ecol. Evol.* **11**, 3956–3976 (2021).
9. Gibbons, S. M., Kearney, S. M., Smillie, C. S. & Alm, E. J. Two dynamic regimes in the human gut microbiome. *PLoS Comput. Biol.* **13**, e1005364 (2017).
10. Poyet, M. *et al.* A library of human gut bacterial isolates paired with longitudinal multiomics data enables mechanistic microbiome research. *Nat. Med.* **25**, 1442–1452 (2019).
11. Wolff, R., Shoemaker, W. & Garud, N. Ecological Stability Emerges at the Level of Strains in the Human Gut Microbiome. *MBio* **14**, e0250222 (2023).
12. Ferrer, M., Chernikova, T. N., Yakimov, M. M., Golyshin, P. N. & Timmis, K. N. Chaperonins govern growth of *Escherichia coli* at low temperatures. *Nat. Biotechnol.* **21**, 1266–1267 (2003).

REVIEWERS' COMMENTS

Reviewer #2 (Remarks to the Author):

I thank the authors for adequately addressing my concerns. I believe this manuscript is ready for publication.

My only comment (which does not require additional review), is that I am still of the opinion that there is no support for claims of doubling time on the order of minutes-to-hours (L17, L48). I think the authors should state this as their own claim (who knows, might turn out to be true) rather than make the appearance that this is a solid fact supported by past studies. I am familiar with those studies and, as mentioned in all my reviews, do not agree with the authors' interpretation.

Reviewer 2

I thank the authors for adequately addressing my concerns. I believe this manuscript is ready for publication.

My only comment (which does not require additional review), is that I am still of the opinion that there is no support for claims of doubling time on the order of minutes-to-hours (L17, L48). I think the authors should state this as their own claim (who knows, might turn out to be true) rather than make the appearance that this is a solid fact supported by past studies. I am familiar with those studies and, as mentioned in all my reviews, do not agree with the authors' interpretation.

Response: We thank the reviewer for their efforts to improve the quality and clarity of our manuscript. We have revised the 'minutes-to-hours' statement at line 48 as follows: "As such, gut bacterial doubling times tend to be fast, likely ranging from minutes-to-hours, although precise *in vivo* estimates are not available (we contend that doubling times of a day or more in the gut would not be sufficient to maintain a stable population size with a daily defecation rate)^{21–23}." We have also added the qualifier 'likely' in front of 'minutes-to-hours' at line 17 in the abstract.